# A spatially-explicit database of wind disturbances in European forests over the period 2000-2018

Giovanni Forzieri[1], Matteo Pecchi[1,2], Marco Girardello[1], Achille Mauri[1], Marcus Klaus[3], Christo Nikolov[4], Marius Rüetschi[5], Barry Gardiner[6,7], Julián Tomaštík[8], David Small[9], Constantin Nistor[10], Donatas Jonikavicius[11], Jonathan Spinoni[1], Luc Feyen[1], Francesca Giannetti[2], Rinaldo Comino[12], Alessandro Wolynski[13], Francesco Pirotti[14], Fabio Maistrelli[15], Savulescu Ionut[16], Stéphanie Wurpillot-Lucas[17], Karlsson Stefan[18], Karolina Zieba-Kulawik[19], Paulina Strejczek-Jazwinska[19], Martin Mokroš[20,21], Stefan Franz[22], Lukas Krejci[23], Ionel Haidu[24], Mats Nilsson[25], Piotr Wezyk[19], Filippo Catani[27], Yi-Ying Chen[28], Sebastiaan Luyssaert[29], Gherardo Chirici[2], Alessandro Cescatti[1], Pieter S.A. Beck[1]

[1] European Commission, Joint Research Centre, Italy
[2] Department of Agriculture, Food, Environment and Forestry, University of Florence, Italy
[3] Department of Forest Ecology and Management, Swedish University of Agricultural Sciences, Umeå, Sweden
[4] National Forest Centre, Forest Research Institute Zvolen, Zvolen Slovakia
[5] Department of Land Change Science, Swiss Federal Institute for Forest, Snow and Landscape Research WSL, Birmensdorf, Switzerland
[6] Institute National de la Recherche Agronomique (INRA), Villenave d'Ornon, France
[7] EFI Planted Forests Facility, 69 Route D'Arcachon, 33612 Cestas, France
[8] Department of Forest Resource Planning and Informatics, Faculty of Forestry, Technical University in Zvolen, Slovakia
[9] Remote Sensing Laboratories, Department of Geography, University of Zurich, Switzerland
[10] Faculty of Geography, University of Bucharest, Romania
[11] Laboratory of Geomatics, Institute of Land Management and Geomatics, Aleksandras Stulginskis University, Kaunas, Lithuania
[12] Regione Autonoma Friuli Venezia Giulia, Direzione centrale risorse agricole, forestali e ittiche, Italy
[13] Provincia autonoma di Trento, Ufficio Pianificazione, Selvicoltura ed Economia forestale, Italy
[14] Department of Land, Environment, Agriculture and Forestry, University of Padua, Italy
[15] Provincia Autonoma di Bolzano, Ufficio pianificazione forestale, Italy
[16] Faculty of Geography, University of Bucharest, Romania
[17] Institut National de l'Information Geographique et Forestiere, IGN, Saint Mandé, France
[18] Swedish Forest Agency, Department of Policy and Analysis, Jonkoping, Sweden
[19] Department of Forest Management, Geomatics and Forest Economics, Institute of Forest Resources Management, Faculty of Forestry, University of Agriculture, Krakow, Poland
[20] Czech University of Life Sciences Prague, Faculty of Forestry and Wood Sciences, Czech Republic
[21] Department of Forest Resource Planning and Informatics, Faculty of Forestry, Technical University in Zvolen, Slovakia
[22] Landesbetrieb Wald und Holz, North Rhine-Westphalia Forest Service, Munster, Germany
[23] Department of Geoinformatics, Faculty of Science, Palacky University, Olomouc, Czech Republic
[24] Laboratoire LOTERR-EA7304, Université de Lorraine, Metz Cedex, France
[25] Southern Swedish Forest Research Centre, Swedish University of Agricultural Sciences, Sweden
[26] Department of Forest Management, Geomatics and Forest Economics, Institute of Forest Resources Management, Faculty of Forestry, University of Agriculture, Krakow, Poland
[27] Department of Earth Sciences, University of Florence, Italy
[28] Academia Sinica, Research Center for Environmental Changes, Nankang, Taipei, Taiwan
[29] Department of Ecological Sciences, Faculty of Earth and Life Sciences, Amsterdam, Netherlands

*Correspondence to*: Giovanni Forzieri (giovanni.forzieri@ec.europa.eu)

**Abstract.** Strong winds may uproot and break trees and represent one of the major natural disturbances for European forests. Wind disturbances have intensified over the last decades globally and are expected to further rise in view of the climate change effects. Despite the importance of such natural disturbances, there are currently no spatially-explicit databases of wind-related impact at Pan-European scale. Here, we present a new database of wind disturbances in European forests (FORWIND). FORWIND comprises more than 80,000 spatially delineated areas in Europe that were disturbed by wind in the period 2000-2018, and describes them in a harmonized and consistent geographical vector format. The database includes all major windstorms that occurred over the observational period (e.g., Gudrun, Kyrill, Klaus, Xhynthia and Vaia) and represents approximately 30% of the reported damaging wind events in Europe. Correlation analyses between the areas in FORWIND and land cover changes retrieved from the Landsat-based Global Forest Change dataset and the MODIS Global Disturbance Index corroborate the robustness of FORWIND. Spearman rank coefficients range between 0.27 and 0.48 (p-value<0.05). When recorded forest areas are rescaled based on their damage degree, correlation increases to 0.54. Wind-damaged growing stock volumes reported in national inventories (FORESTORM dataset) are generally higher than analogous metrics provided by FORWIND in combination with satellite-based biomass and country-scale statistics of growing stock volume. The potential of FORWIND is explored for a range of challenging topics and scientific fields, including scaling relations of wind damage, forest vulnerability modelling, remote sensing monitoring of forest disturbance, representation of uprooting and breakage of trees in large-scale land surface models and hydrogeological risks following wind damage. Overall, FORWIND represents an essential and open-access spatial source that can be used to improve the understanding, detection and prediction of wind disturbances and the consequent impacts on forest ecosystems and the land-atmosphere system. Data sharing is encouraged in order to continuously update and improve FORWIND. The dataset is available at https://doi.org/10.6084/m9.figshare.9555008 (Forzieri et al., 2019).

## 1 Introduction

Natural forest disturbances represent a serious peril for maintaining productive forests. Studies indicate that their occurrence can reduce primary production and partially offset carbon sinks or even turn forest ecosystems into carbon sources (Kurz et al., 2008; Yamanoi et al., 2015; Ziemblińska et al., 2018). This is particularly critical for windthrow and tree breakage due to strong winds, which represent one of the major natural disturbance for European forests (Schelhaas et al., 2003; Seidl et al., 2017). Such disturbances are intensifying globally, a trend which is expected to continue with further climate change (Bender et al., 2010; Knutson et al., 2010; Seidl et al., 2014).

European windstorms are associated with areas of low atmospheric pressure that typically occur in the autumn and winter months (Martínez-Alvarado et al., 2012). Deep low-pressure areas frequently track across the North Atlantic Ocean towards Western Europe, pass the north coast of Great Britain and Ireland and into the Norwegian Sea. However, when they track further south, they can potentially hit any country in Europe. In 1999, storm Lothar damaged approximately 165 million $m^3$ of timber mainly in France, Germany and Switzerland (Gardiner et al., 2010), which is equivalent to about 140% of the average

annual round-wood harvested in the countries affected (FAOSTAT, 2019). In 2005, 75 million m³ were damaged by storm
Gudrun in Sweden (Gardiner et al., 2010), equivalent to about one year's cuttings in the same area (FAOSTAT, 2019). In 2007, the storm Kyrill caused the loss of 49 million m³ of timber in Germany and the Czech Republic. In 2009 and 2010, storms Klaus and Xynthia hit forests in France and Spain and caused timber losses totalling approximately 45 million m³. In 2018, the Vaia storm hits the North-Eastern regions of Italy causing a damaged growing stock volume of about 8.5 million m³. The socio-economic consequences of wind disturbances can be critical especially for local economies highly dependent on the
forest sector. Countries in Northern Europe and Central-Eastern Europe, where the forest sector may cover up to 6% of the national GDP (FOREST EUROPE, 2015), are, therefore, potentially more vulnerable to wind-related impacts.

Despite the risks they pose, spatially explicit databases of wind disturbances across Europe currently do not exist. Recent assessments of current and future forest damages due to windstorms at European scale are based on catalogues of disturbances collected at country level (Gregow et al., 2017; Schelhaas et al., 2003; Seidl et al., 2014). Such databases (e.g., FORESTORM)
are subject to multiple sources of bias and uncertainty associated to the diversity of the underlying inventories. Furthermore, estimates of forest damage aggregated at national scale may only partially represent the spatial variability of the phenomenon. In fact, the coarse spatial resolution of such data hampers inferential analysis of potential drivers of forest vulnerability and their use in spatially explicit models to monitor or forecast wind-related impacts (Masek et al., 2015; Phiri and Morgenroth, 2017). Despite the lack of systematic mapping of wind disturbances in European forests, a multitude of local, national, and
transnational initiatives have accurately mapped forest areas affected by wind over the last decades. These data represent highly informative observational records to characterize spatial patterns of forest damages. However, they are collected by different institutes, and are often difficult to retrieve or poorly documented. Since 2012, the Copernicus Emergency Management Service (https://emergency.copernicus.eu/) produces maps of natural disasters throughout the world based on the analysis of satellite images and other geospatial data. While this important initiative can help map wind-affected areas, it only covers
recent years and, being an on-demand service, it is not comprehensive as it depends on the interests of individual authorized users of the service to map a given forest disturbance.

In this study, we try to fill the above-mentioned gap. To this aim, we collected and harmonized 89,743 forest areas damaged by wind into a consistent geospatial dataset. The work was carried out through a unique joint effort of 28 research institutes and forestry services across Europe. This collaboration led to the first spatially-explicit database of wind disturbances in
European forests over the period 2000-2018, hereafter referred to as the FORWIND database. We believe that it provides essential spatial information to improve our understanding of forest damage from wind and can assist in large-scale systematic monitoring and modelling of forest disturbances and their effects on the land-atmosphere system. In the following sections, we describe the data collection, the harmonization process, and the cross-comparison performed against satellite-retrievals of changes in vegetation cover and data from national inventories of forest disturbances. We conclude the data description with
some examples of the possible usage of the FORWIND database.

**2 Methods**

We collected wind disturbances events caused by windstorms or tornadoes that occurred in Europe between 2000 and 2018. A wind disturbance event is represented by a georeferenced polygon that delineates the damaged forest stand, regardless of the degree of damage. The original acquisition of the polygons was made by aerial/satellite photointerpretation or field survey (Table 1). Therefore, the polygons are delineated when a reasonably homogeneous patch of damaged forest is detected from the ground or remotely. The data were managed mostly on the Google Earth Engine platform (Gorelick et al., 2017) to efficiently quantify the extent of disturbances over large scales and extract additional informative attributes (e.g., Hansen et al., 2013; McDowell et al., 2015). We structured the data collection process in four main phases, described below.

- **Literature review and data gathering.** We searched PubMed and Scopus for articles published up to January 2019, with no language restrictions, using the search terms "wind disturbance" OR "windthrow" OR "forest damage" OR "wind damage" OR "forest disturbance" AND "Europe" OR single country name in the publication title OR abstract. The identified studies had mainly mapped the effects of wind on forests for single events and/or for a limited areal extent. We then retrieved the spatial delineation of the observed wind damages from the corresponding authors or contact persons responsible for the data acquisition. The collected data were originally recorded by different research institutes and international initiatives across Europe using diverse methodologies. Table 1 lists the data providers and the acquisition methods.

- **Coordinate system transformation.** The wind disturbances were transformed to the same geographical unprojected coordinate system (World Geodetic System 1984, WGS84, EPSG:4326).

- **Spatial segregation.** The spatial segregation of each record was verified. In case multiple features for the same event overlapped, they were merged.

- **Harmonization of the degree of damage.** A damage classification for forest disturbances was originally recorded for windstorms that occurred in France in 2009, in Lithuania in 2010, in Germany in 2017, in Italy in 2015 and –for part of the records - in 2018. In order to make these records comparable in terms of the severity of damage, the original classes were harmonized into a single damage metric following the rationale reported in Table 2. The resulting degree of damage varies between 0 (no damage) and 1 (full destruction of the forest patch). Information on the degree of damage is available for ~48% of records and is included as a basic attribute when available (Table 3).

Table 1

Table 2

Table 3

## 3. Data records

The FORWIND database is the final output of the data collection procedure and it is publicly available at
https://doi.org/10.6084/m9.figshare.9555008 (Forzieri et al., 2019). The FORWIND dataset contains records as polygon features in shapefile format (.shp). The geometry of a feature is stored as a shape comprising a set of vector coordinates corresponding to the boundaries of the area of a given wind disturbance. Records are georeferenced in geographical coordinates, i.e. latitude and longitude, following the WGS84 standard (EPSG:4326). Basic attributes of each disturbance (Table 3) are provided in an associated table, stored in a .dbf file.

Overall, FORWIND includes 89,743 records, corresponding to ~1 million ha of forest area affected by wind disturbances during the 2000-2018 period. Each record should not be viewed as independent as a single storm may cause multiple, geographically disjunct, disturbances. At European level, the median wind-caused forest disturbance patch measures 1.07 ha (Table 4). However, there is substantial variability across disturbances and countries likely driven by the high heterogeneity of forest and landscape characteristics. Figure 1 shows the spatial and temporal variations of records in the FORWIND database. In order to better visualize the data, we summed the areas affected by wind disturbances in 0.5-degree cells (Fig. 1a). A similar aggregation was used to show the timing of the disturbances, here expressed as the year in which most area was disturbed within a given cell (Fig. 1b). The current release of FORWIND includes wind disturbances that occurred in Austria, Switzerland, the Czech Republic, France, Germany, Ireland, Italy, Lithuania, Poland, Romania, Russia, Slovakia and Sweden. The major windstorms that occurred in the last two decades are included in FORWIND, particularly Gudrun in 2005 (Sweden), Kyrill (Germany) in 2007, Klaus in 2009 (France), Xhynthia in 2010 (Germany) and Vaia in 2018 (Italy). The high spatial detail of FORWIND is illustrated in Figure 2 for some key windstorms. According to the institutions responsible for the data acquisition, the wind disturbances recorded in FORWIND exhaustively represent the damaged forest areas caused by those specific events. However, some known damaging wind events are currently missing in the database. In order to provide a more comprehensive assessment of the representativeness of FORWIND, we derived for each country the ratio between the number of wind events included and the number of all wind events that occurred and which are known to have caused forest damages (Table 5). The number of known damaging events is derived by summing up the number of distinct events recorded in FORESTORM (http://www.iefc.net/storm/) and FORWIND during the 2000-2018 period. Therefore, the temporal representativeness ranges between 0 (all known wind disturbances are missing in FORWIND) and 1 (all known wind disturbances are included in FORWIND). Estimates of representativeness ranges between 0.13 and 1 amongst the countries included in FORWIND, with an average value of 0.63 at the European level (see Table 5). However, when countries currently missing in FORWIND are also accounted for, the average representativeness decreases to 0.30. These values should be viewed with caution as the estimated number of total damaging wind events resulting from FORWIND and FORESTORM could likely deviate from the actual ones. Future efforts should be aimed at populating FORWIND with the damaging wind events known to be missing.

## 4. Comparison of FORWIND with satellite-based metrics and national inventories

The lack of alternative datasets with the same spatially explicit mapping of wind disturbances as in FORWIND does not allow for a standard validation exercise. Therefore, we evaluated the validity of FORWIND based on the plausibility of the collected spatial delineations of wind disturbances with respect to two satellite-based proxies of forest disturbances and estimates of

185 forest damages reported in national inventories.

### 4.1 FORWIND versus LANDSAT-based forest cover loss

FORWIND was initially compared with satellite-based estimates of forest cover loss derived from the Global Forest Change maps (Hansen et al., 2013) (GFC, https://earthenginepartners.appspot.com/science-2013-global-forest). GFC maps characterize the annual forest coverage at global scale during the period 2000–2018 at 30-meter spatial resolution based on

time-series analysis of Landsat images. Forest cover loss is defined as an area that has changed from a state of forest to non-forest, following a given disturbance event (natural or anthropogenic). The change detection is based on the variation in the spectral properties of the land surface. Windstorm events in Europe often occur in autumn and the beginning of winter, when the availability of cloud-free images is typically much more limited than in summer. Hence, satellite retrievals of forest cover loss may miss the exact timing of the disturbance. Therefore, the GFC-based forest cover loss may only record wind

disturbances the year after the event occurred. In addition, fallen trees following a windstorm or tornado often maintain their leaves for months. This may lead to limited or no change in land reflectance properties, even when cloud-free images are available. Therefore, satellite-based products may underestimate forest cover loss in the short-term (interannual scale). In order to account for these effects, we considered the forest cover loss by summing up the forest loss over the year of a given event together with that of the following year (lag-01). The loss estimate was quantified with respect to the pre-event conditions (the

forest cover in the year before the event). To reduce potential contamination effects from other disturbances on the resulting total forest cover loss, we removed areas affected by fires the year following a wind event. Information on forest areas affected by fires were retrieved from the European Forest Fire Information System (EFFIS, http://effis.jrc.ec.europa.eu/). Insect outbreaks, which may be triggered by large numbers of dead trees following wind disturbances (Stadelmann et al., 2013), generally lead to a slow change in tree cover, which may only marginally affect the 1-year temporal lag used for our estimates

of forest cover loss. Furthermore, forest logging following a wind event can be considered a secondary effect of the strong winds, as it is often employed to reduce the risk of other forest disturbances (specifically insect outbreaks and fires). Therefore, the resulting estimates of forest cover loss for the selected areas should reflect wind disturbances first and foremost. We

emphasize that Landsat-derived estimates of forest cover loss are affected by the uncertainty in satellite retrievals and do not represent the true impacts. However, their suitability for detecting forest disturbances over large scale has been widely recognized (Curtis et al., 2018; Hansen et al., 2013) and, therefore, they are here considered a good proxy of forest loss.

For each selected FORWIND record we computed the area of affected forest based on the spatial delineation of the polygon and the corresponding Landsat-derived forest cover loss and calculated the correlation between the two sets of estimates. In order to account for the spatial dependence structure of FORWIND data, correlation values were derived for 100 subsets of 1000 records randomly selected from the entire dataset. The final estimate of correlation was then quantified as the average of the correlation values derived from the 100 subsets.

Results for the whole dataset are shown in Figure 3a. Overall, we found a modest but significant Spearman rank correlation coefficient ($\rho_k$=0.48, p-value<$10^{-3}$), which supports the validity of FORWIND in mapping areas subject to changes of forest coverage due to wind disturbances. We point out that for this calculation we did not mask the data based on the degree of damage, because such information is available only in some countries. However, a similar correlation analysis performed by rescaling the recorded areas based in their damage degree (for those records that report the information) led to higher correlation values up to 0.54. We further tested the sensitivity of our results to the temporal lag used to quantify the forest cover loss. To this aim, we complemented the previous analysis (lag-01) using Landsat-based forest cover loss estimated for the year of the event only (lag-0) and the following year only (lag-1). In order to investigate possible scaling relations, the correlation analysis was performed accounting for the FORWIND records with a spatial extent above a given threshold derived from the percentiles 0, 0.25, 0.50 and 0.75 of the full dataset (corresponding to about 0, 0.5, 1, and 3.5 ha, respectively). Results show that correlation values between FORWIND affected areas and lag-0 forest cover loss tends to slightly decrease with an increasing size of the wind disturbance (Fig. 3b). The opposite pattern is observed for correlation values with lag-1 forest cover loss. The forest cover loss accumulated over the two years considered (lag-01) appears dominated by the contribution of lag-1 forest cover loss. We argue that such contrasting tendencies may be linked to the scale and climatology of extreme winds. Wind-related forest impacts of limited areal extent originate from local windstorms or tornadoes that may occur throughout the year. For these events, most of the damage is probably well captured by lag-0 effects, as it is more likely that cloud-free images are available after the event. In contrast, the larger and more damaging windstorms, which affect larger forest areas, typically occur in autumn and early winter (decreasing the likelihood of cloud-free images after the storm and before the end of the year). For these events, the inclusion of the lag-1 effect is key to characterize the impact on forest cover.

Figure 3

## 4.2 FORWIND versus MODIS Global Disturbance Index

FORWIND was also compared with an independent dataset of satellite-based estimates of forest disturbance as expressed by the MODIS-based Global Disturbance Index (Mildrexler et al., 2009) (MGDI,

). MGDI maps quantify the overall annual forest disturbance globally for the period 2004-2012 at 500-meter spatial resolution. The disturbance retrieval is based on the variations in the Enhanced Vegetation Index and land surface temperature following a given sudden change in forest cover. Consistent with the previous Landsat-based analysis - the total change in MGDI potentially related to a given wind disturbance was computed as the accumulated net change in MGDI over the event year and the following year (lag-01). The change was quantified with respect to the pre-event conditions (MGDI in the year before the event). The technique used to disentangle the fire signal, as well as the correlation and sensitivity analyses with respect to the temporal lags and wind disturbance size, were performed analogously to the previous validation exercise (Section 4.1).

Overall, we found a low but significant correlation coefficient ($\rho_k$=0.27, p-value<$10^{-3}$) (Fig. 3c). The lower correlation compared to the Landsat-based dataset is presumably due to the coarser spatial resolution of MGDI that probably does not fully capture the changes in land surface properties due to wind disturbances (Mildrexler et al., 2009). This seems to be supported by the generally increasing correlation values up to 0.31 for wind disturbances of 1 ha consistently across the different temporal lags (Fig. 3d).

## 4.3 FORWIND versus FORESTORM

FORWIND data were finally compared with estimates of damaged growing stock volume (GSV) that are recorded at country level in the FORESTORM database for five windstorm events: Slovakia in 2004; Sweden in 2005 (Gudrun storm), Germany in 2007 (Kyrill storm), the Czech Republic in 2007 (Kyrill storm) and France in 2009 (Klaus storm). We derived the damaged GSV by multiplying the estimated GSV by the percentage damaged, both of which are reported in FORESTORM. An analogous metric was derived from FORWIND data by first calculating for each FORWIND record the amount of GSV lost by multiplying the areal average GSV by the damage level reported for the record. As the damage level was only reported for Klaus, for the other events we assumed a damage level equal to the average level reported for Klaus weighted on the spatial extent of each record. The GSV was retrieved from the GlobBiomass dataset (Santoro et al., 2018) (https://doi.pangaea.de/10.1594/PANGAEA.894711) which is based on multiple remote sensing products and is considered the state-of-the-art global biomass product. This satellite-based GSV estimate refers to the year 2010 and has a spatial resolution of 100 meter. The damages to GSV were then summed by event and country. Event-scale FORWIND damaged GSVs were then compared with estimates derived from FORESTORM.

Overall, results show that the magnitude of damages estimated from FORWIND and FORESTORM are largely different, except for the 2009 Klaus storm in France for which we found a very good agreement (Fig. 3e). For most of the events, however, FORESTORM tends to systematically give higher forest damage estimates than FORWIND with differences exceeding 90%. We note that such differences persist when we derive FORWIND estimates of damaged GSV assuming a 100% damage degree for all records (not shown). Therefore, the uncertainty in the damage degree in FORWIND does not affect substantially the difference between FORWIND and FORESTORM. We recognize that estimates of forest damages based on FORWIND are fully dependent on the GSV derived from GlobBiomass. Indeed, any deviations of the mapped GSV

from the true forest state are inherently translated into our damaged GSV estimates. In particular, the GSV map refers to the year 2010, therefore it is very likely that it largely reflects the biomass conditions following, rather than preceding, the windstorm events (all the five events considered in this validation exercise occurred before 2010).

In order to disentangle such source of bias we derived country-scale estimates of average GSVs for the year 2000 (pre-event conditions) from the State of Europe's Forest (FOREST EUROPE, 2015) (https://www.foresteurope.org/docs/SoeF2015/OUTPUTTABLES.pdf). We then derived the damaged GSVs by multiplying Forest Europe-derived GSVs by the total forest area affected for each of the considered wind events by assuming a 100% degree of damage. Furthermore, as wind disturbance typically affects taller forest patches and probably more productive trees compared to the country scale average, we rescaled previous estimates of damaged GSVs based on the ratio between the average tree height computed over wind-affected areas and the average tree height computed over the whole vegetated land in the country. Tree height values where retrieved from 1-km spaceborne light detection and ranging (lidar) data acquired in 2005 by the Geoscience Laser Altimeter System (GLAS) aboard ICESat (Ice, Cloud, and land Elevation Satellite), (https://webmap.ornl.gov/wcsdown/dataset.jsp?ds_id=10023) (Simard et al., 2011).

Similar to the previous results, except for the Klaus storm, we found higher values of damaged GSVs in FORESTORM than in our estimates based on the integration of FORWIND and country values of GSVs (Fig. 3f). We recognize that FORWIND could miss some wind damage occurrences, for instance due to incomplete detection of wind disturbance from aerial photointerpretation or difficulties of mapping inaccessible areas by ground surveys. However, according to the institutions responsible for the data acquisition, the forest areas affected by the windstorm events considered in this validation exercise were exhaustively mapped. Therefore, possible residual omissions are expected to only marginally affect our results. We therefore argue that a possible source of error may be associated to the FORESTORM database. Estimates of forest damages from FORESTORM originate from different sources and are collected by multiple actors. Hence, the loss figures should be viewed in light of their potential biases, including a possible overestimation of the true impacts.

# 5 Possible applications of FORWIND database

For demonstration purposes, we show a series of possible applications of the FORWIND database. We recognize that the examples described in the following sections are an oversimplification of the relationships observed in nature and of the biomechanical processes that may cause wind disturbances or that can be triggered by wind disturbances. More sophisticated approaches could be employed to better explore and predict the forest response functions to wind disturbances. For example, multiple variables, susceptibility factors, and drivers (e.g., tree species, tree dimension, management regimes, planting patterns, soil depth, snow cover), contribute concurrently to modulate the forest response to wind disturbances (Hart et al., 2019; Klaus et al., 2011; Mitchell, 2013) and their contribution should be analysed in a multidimensional space (e.g., Section 5.1 and 5.2).

Therefore, the approaches described here should not be considered as a reference methodology but only as informative applications to explore the usefulness of the FORWIND database.

## 5.1 Scaling relations of severity of wind disturbances

The exploration of the relations between forest dynamics and scale can reveal important information on ecosystem spatial organization by addressing preservation of information integrity in upscaling/downscaling procedures of land-surface
parameterization for ecological modelling applications (Forzieri and Catani, 2011). Here, we explore – in a simplified approach – the scaling relations of the degree of damage of wind disturbances collected in FORWIND. To this aim, we estimated, for each record, the cover fractions of different plant functional types (PFTs) including broadleaf deciduous (BrDe), broadleaf evergreen (BrEv), needleleaf deciduous (NeDe) and needleleaf evergreen (NeEv). Cover fractions were retrieved from the annual land cover maps of the European Space Agency's Climate Change Initiative (ESA-CCI, https://www.esa-landcover-
cci.org/). The degree of damage of each record was then spatially averaged over the sampled interquartile range of affected areas (bin size of 0.25 ha). The spatial averages were computed separately for each PFTs utilizing their cover fractions as weights. Quadratic polynomial functions were finally used to fit the observations and retrieve the relationship between the degree of damage and affected area for the considered PFTs.

Results show that all considered PFTs generally have a higher degree of damage for wind disturbances with small spatial
extent (Fig. 4a). This may reflect a better delineation of small affected areas when the damage is typically higher and homogeneous. Furthermore, the declining scaling relations suggests potential spatially-varying dampening effects of wind severity due to landscape heterogeneity over large areas compared to more homogeneous patterns in small forest patches. Model fitting shows reasonably good performances with $R^2$ ranging between 0.84 and 0.90 across the PFTs (Table 6). Compared to the other PFTs, NeEv generally has a higher degree of damage that is related to the affected area by a quasi-
monotonic pattern. The relationships found for the other PFTs show a stronger link between the degree of damage and affected area compared to NeEv, particularly over the range with larger samples (affected areas < 2 ha, Fig. 4b) as visualized by the steeper slopes of the fitting functions. For BrDe, BrEv and NeDe a prominent parabolic pattern emerges distinctly driven by records with a large spatial extent and a relatively high degree of damage.

Table 6

Figure 4

## 5.2 Forest vulnerability to wind disturbances

The vulnerability of forests to natural disturbances is a key determinant of risk and reflects the propensity of a forest to be
adversely affected when exposed to hazardous events (IPCC, 2014). Vulnerability is largely controlled by local environmental conditions, such as climate and forest characteristics, which regulate the sensitivity of ecological processes to disturbance

agents (Lindenmayer et al., 2011; Seidl et al., 2016; Turner, 2010). Here, we employ FORWIND records to quantify the forest vulnerability as a function of the fraction of evergreen needleleaf forest and annual maximum wind speed. The fraction of NeEv was derived from the ESA-CCI product aggregated at 0.5 degree spatial resolution. Annual maximum wind speeds were computed from NCEP/NCAR Reanalysis 2 data (Saha et al., 2010) (NCEP2, https://www.esrl.noaa.gov/psd/data/gridded/data.ncep.reanalysis2.html). Daily average wind data at 0.5 degree spatial resolution were acquired and the two horizontal components combined to derive the magnitude of the wind vector. For each cell, the fraction of NeEv and the annual maximum wind concomitant with a wind disturbance were then selected from the time series and used in our experiment as potential drivers of vulnerability (Fig. 5a,c). The values of fraction of NeEv and annual maximum wind speed (predictors) were linked with the corresponding FORWIND affected area (response variable) within each 0.5 degree cell. The high spatial variability of the considered metrics and the potential effects of additional environmental factors not considered in this exercise may potentially mask the functional relations between the response variable and predictors. In order to reduce such potential sources of noise, response variables and predictors were spatially averaged over the sampled range of the predictors (bin sizes of 10% and 2 m/s for fraction of NeEv and annual maximum wind speed, respectively).

Wind disturbance areas manifest a substantial variability, as evident form the generally high values of the coefficient of variation. However, when data are spatially averaged at bin level, simple linear regression models show a reasonably good fit, with $R^2$ values of 0.52 and 0.81 for the fraction of NeEv and annual maximum wind speed, respectively. Emerging patterns are largely consistent with expectations and previous studies. An increasing fraction of NeEv leads to an increase in wind disturbance area (growing rate of 12 ha of affected forest per 0.1 increase in NeEv fraction, Fig. 5b). The emerging relation is likely driven by the relatively high abundance of *picea abies* in the sampled forest areas. This tree species is typically characterized by shallower rooting systems often due to the type of soils on which it is planted (Mason and Valinger, 2013). Combined with the limited flexibility of its branches (Mayhead, 1973) and relatively low rupture strength of its trunk (Lavers, 1969) this makes *picea abies* prone to uprooting and breakage by strong winds (Colin et al., 2009; Nicoll et al., 2006). A similar pattern emerges with respect to annual maximum wind speed (Seidl et al., 2011). Wind disturbance area tends to increase with rising wind speed (growing rate of 32 ha of affected forest per 1 ms$^{-1}$ increase in wind speed, Fig. 5d). Maximum wind speeds are the primary determinant of wind disturbances. However, we point out that the coarse spatial and temporal resolution on NCEP2 data largely underestimate the speed of wind gusts and may completely miss peak winds originating from tornados. This is clearly evident from the range of values of annual maximum wind speed (6-22 m/s) which are far lower than the wind speeds reported in country-scale inventories of forest disturbance (e.g., 42 m/s for Gudrun, FORESTORM) and in the Extreme Wind Storms (XWS) catalogue (Roberts et al., 2014) (http://www.europeanwindstorms.org/) (e.g., 39 m/s for Gudrun, XWS).

Figure 5

## 5.3 Remote sensing detection and attribution of wind disturbances

Natural disturbances are accelerating globally, but their full impact is not quantified because we lack an adequate monitoring system. Remote sensing offers a means to quantify the frequency and extent of disturbances over landscape-to-global scales (McDowell et al., 2015). For instance, some pioneering studies have begun producing classification maps of various forest disturbance agents based on remote sensing data (Cohen et al., 2016; Hermosilla et al., 2015; Potapov et al., 2015; White et al., 2017). However, the attribution of forest change to windstorms remains challenging. Previous systematic monitoring has been performed only over limited areal extents and showed considerable uncertainty (Baumann et al., 2014; Schroeder et al., 2017) mostly due to the limited number of sampled wind-affected areas available for training/testing classification algorithms (Schroeder et al., 2017). In this respect, FORWIND data can be used to enhance our capability to detect and attribute forest damage due to windstorms from remote sensing data. Here, we tested different types of classification trees in combination with a Sentinel-2 imagery and FORWIND database to automatically map wind disturbances that occurred following storm Vaia in October 2018 in the Dolomites Mountains in Northern Italy (Pirotti et al., 2016). Google Earth Engine was used to create a single image composite from a stack of cloud-free pixels (11 and 28 images acquired before and after the windstorm event, respectively). Median was used as a reducer over the vector of pixel values derived from each image, after masking cloudy pixels using the cloud probability raster delivered from atmospheric, terrain and cirrus correction of the sen2cor processor (Louis et al., 2018). Further masking was applied to process only pixels covered by forest, using the 2018 estimated forest cover map from the Global Forest Change 2000–2018 dataset (Hansen et al., 2013). Binary classification, i.e. damaged vs. non-damaged, was applied over a set of 1000 completely damaged areas retrieved from FORWIND, and 1000 non-damaged areas. Half of these were used for training and validation, the other half for unbiased testing of the model performance. The feature vector used for predictors included reflectance values recorded by Sentinel-2 after radiometric and atmospheric correction (i.e. bottom of atmosphere) and a tasselled cap (TC) transform of reflectance bands to the brightness, greenness and wetness domain. The TC was added as it is reasonable that wind-affected areas will provide higher degree of brightness and lower degree of greenness with respect to undisturbed areas (Baumann et al., 2014). Several machine learning algorithms were employed, including Random Forest, Extremely-Randomized Forest, Gradient Boosting Machines, Deep Neural Networks and Stacked Ensemble, all trained and cross-validated based on K-fold validation with K=5 (Click et al., 2016).

Results, based on the best performing classification model (Random Forest), provided very promising accuracy with a F1 score of 0.97, with 27 false positives and 1 false negative over 915 pixels used for testing (507 not-damaged and 408 damaged). Figure 6 shows mapped probability of wind occurrence - with blue to red respectively representing zero to one probability of a heavily hit area in the Veneto Region. Based on visual comparison with ground data, the automatic classification is able to capture the spatial patterns of wind damage. It is worth noting that damage in forest/non-forest nexus is less accurate due to pixel mixing. Another point worth further investigation is what might be defined as false positives from binary classification, might actually be true positives that were not mapped due to human error. On the other hand, false negatives might be true

negatives in the sense that small patches of standing trees might be present in mapped areas due to the understandable minimum level of detail that must be adopted.


Figure 6

## 5.4 Representation of wind disturbances in Land Surface Models

Land surface models (LSM) are key components of Earth System Models that are widely applied to support policy-relevant
assessments on the impact of climate change on terrestrial ecosystems (Quéré et al., 2018). Recently, windstorm effects have been incorporated in LSMs (Chen et al., 2018). However, these models are hampered by the lack of harmonized spatially-explicit information on windstorms required as input for robust model parameterization and large-scale representation of wind disturbance. In such contexts, the FORWIND database represents a valuable source of harmonized wind-affected forest areas for improving model calibration and/or evaluation. To illustrate such possible application, FORWIND was used as an
independent data source to evaluate the LSM ORCHIDEE (revision r4262) that simulates windthrow damages and that was parameterized with observation prior to the FORWIND time frame.

ORCHIDEE r4262 was parameterized to the extent possible with observed parameter values. Nevertheless, tuning windthrow parameters remained necessary for gustiness, maximum damage rate (which is a parameter to account for the large simulations units, i.e., 2500 km$^2$, in ORCHIDEE vs. the small scale at which storm damage occurs), and the relaxation factor for the
damage function (Rf in eq.(12) in (Chen et al., 2018); which is the parameter that converts the difference between the critical and actual wind speed into a damage rate). To this aim Swedish data from 1981 to 2000 (Nilsson et al., 2004), a period characterized by the absence of major storms in Sweden, was selected. Tuned parameters reproduced the annual storm damage in Sweden between 1981 to 2000 with a root mean square error of 1.3 Mm$^3$ year[-1] as well as the observed damage from the 2005 storm named Gudrun (75 Mm$^3$ of reported damage vs. 77 Mm$^3$ of simulated damage) (Chen et al., 2018). Subsequently,
the parameter values obtained by tuning ORCHIDEE against the damage rate in the absence of major storms in Sweden were used to simulate windthrow over the entire European domain starting in the year 2000.

The model simulated a total annual damage of 30 Mm$^3$ year[-1] of wood timber over an area of 2Mkm$^2$ averaging 0.15 m$^3$/ha/year which is in line with the reported value of 0.13 m$^3$/ha/year between 1951 and 2000 (Schelhaas et al., 2003) and the projected 0.15 m$^3$/ha/year-1 between 2000 and 2020 (Seidl et al., 2014). According to ORCHIDEE, storms affected a total of 50,000
km$^2$ between 2000 and 2015, where, damage area was obtained by dividing the damaged timber volume (m$^3$ m$^{-2}$) by the sum of the damaged and remaining timber volume (m$^3$ m$^{-2}$) and multiplying by pixel surface area. At first sight these results strongly contrast with the 14,000 km$^2$ of storm damaged area archived in the FORWIND database between 2000 and 2015 but it should be noted that FORWIND was estimated to represent just 30% of the European storms since 2000 (see Table 5). Extrapolating FORWIND to the European domain suggests that based on the observations, the area affected by wind storms could exceed
38,000 km$^2$.

Differences in spatial and temporal definitions between ORCHIDEE and FORWIND were partly accounted for by extracting storm damage estimates from ORCHIDEE only when the storm was included in FORWIND. Following this, the ORCHIDEE model appears to overestimate the damage rate in years with small storms but failed to estimate the damage rate of Klaus in 2009 (Fig. 7). This suggests that the tuned relaxation factor for the damage function (Rf=6), which allows for individual tree

damage at actual wind speeds below the critical wind speed, is too high. As a consequence ORCHIDEE simulates too much small-scale damage at wind speeds below the critical value, while the maximum damage rate in ORCHIDEE is too low. Furthermore, ORCHIDEE could only partially represent the effects of forest stand edges on the propagation of wind disturbance. Indeed, damage due to the Klaus storm was particularly amplified due to the amount of damage arising at vulnerable forest stand edges and then propagating through the uniform *pinus radiata* stands (Hart et al., 2019; Kamimura et

al., 2015).

These results shows that evaluating the capacity of land surface models to project storm damage hinges on our ability to precisely define the storm events recorded in the databases and our ability to use this information to estimate key model parameters such as the relaxation factor and the maximum damage rate.

Figure 7

## 5.5 Indirect effects of wind damages on slope instability

FORWIND may also be employed to improve the predictive performances of slope stability models that rely on water-soil interactions and soil mechanics. Vegetation affects terrain properties in a variety of ways including the modification of

hydraulic conductivity, the regulation of evapotranspiration and the increase of soil strength by apparent root cohesion (Amundson et al., 2015; De Baets et al., 2008). This, in turn, may strongly condition terrain response to external forcing such as intense rainfall and seismic shaking, leading to mass wasting in the form of shallow landslides and soil erosion (Moos et al., 2016; Ruiz-Colmenero et al., 2013).

We have tested the capability of FORWIND to provide data for assimilation in shallow landslide hazard models and for model

validation by selecting the dataset relative to the Vaia wind storm of October 2018 in the Dolomites Mountains in Northern Italy and using it to model indirect effects of wind disturbance on slope stability. A multivariate machine learning model for shallow landslide susceptibility has been trained and applied on pre-storm terrain attributes to reveal relative probability of occurrence and then applied again to post-storm conditions to measure the effects of forest disturbance on the hazard. The terrain attributes considered in the analysis include elevation, slope angle, slope curvature variability, local rainfall patterns,

geo-mechanical classes, potential soil saturation, contributing area and pre- and post-storm Normalized Difference Vegetation Index (NDVI) maps from Landsat 8 level-2 imagery. The dataset was trained by a RUSBoosted Random Forest regressor (Catani et al., 2013) on a validated shallow-landslide dataset derived from the Italian National catalogue IFFI (Trigila et al., 2013). The training process highlights that NDVI, typically considered as a good proxy of biomass density, is ranked second

in terms of explained variance and seems to strongly condition landslide susceptibility in all the Dolomites Mountains. The
FORWIND database collects dated and graded information on wind damage to forests that directly correlates to marked
changes in NDVI values, as can be observed in Fig. 8a. The effects of the damages recorded in the FORWIND dataset are
measurable by comparing the levels of susceptibility before and after the occurrence of the Vaia wind storm (Fig. 8b). As can
be appreciated in the map, the red areas, that reveal a marked increase in the probability of landslides, match the FORWIND
polygons very well and clearly indicate the usefulness of the wind-damage geographical databases in slope hazard prediction
and modelling. In Fig. 8b we also note some omission and commission errors. They, however, can be easily explained by
noting that vegetation stripping (or vegetation scantiness) is only one of the factors contributing to landslides. Therefore,
wherever Vaia has damaged forests but slopes are very gentle, no shallow landslides can be generated. On the other hand,
outside FORWIND polygons landslides may still develop, due to the prevailing action of other factors, such as e.g.
unfavourable geological conditions or strong concentrated rainfall.

The use of FORWIND data in landslide modelling is not limited to the cross-validation of biomass volume changes but can
also be extended to the usage of the dataset as an additional predictor in multi-variate statistics. We noted that the overlapping
of FORWIND polygons and NDVI stress (brown) areas shows few exceptions. In such areas, the two factors seem to behave
independently. In particular, locations where wind damage do not correspond to a NDVI change might reveal cases where the
possible storm effects on soil stability are not captured by satellite-based variations in biomass content and must be accounted
for by a different metric. That, in turn, opens the way to important future developments in the usage of wind-driven damage
datasets in slope stability forecasting.

Figure 8

## 6 Conclusions


Modern and forthcoming Earth observation systems (McDowell et al., 2015), new generation of Land Surface Models (Bonan
and Doney, 2018), recent developments of cloud computing platforms (Gorelick et al., 2017) and machine learning approaches
(Reichstein et al., 2019) are offering unprecedented opportunities to explore and predict ecosystem dynamics at an increasing
spatial-temporal resolution and sophistication level. In light of such progress, it is of paramount importance to implement
robust calibration and validation procedures based on reliable ground observations. In order to capture the variability of
ecosystem response across wide environmental gradients, reference ground truth needs to be collected over large spatial scales.
In this context, FORWIND represents an essential dataset to improve our capacity to understand, detect and predict wind
disturbances and quantify their impact on forest ecosystems and the land-atmosphere system. The FORWIND database is the
first Pan-European collection of spatially delineated forest areas affected by wind disturbances and includes all major events
that occurred over the 2000-2018 period. Future research needs should be aimed at further populating FORWIND with missing
damaging wind events.

## 7 Data availability

Data are freely available at https://doi.org/10.6084/m9.figshare.9555008 (Forzieri et al., 2019a) and will be periodically
updated with new and historical events. To this effect, the authors welcome further data contributions and commit to properly
acknowledging them.

**Author contributions.** G.F. designed the study. M.P. performed the data harmonization. M.G. assisted in data integration
tasks, M.M., C.Nikolov., M.R., J.T., D.S., C.Nistor., D.J., B.G., F.G., R.C., A.W., F.P., F.M., S.I., W.L-S., K.S., K.Z-K., P.S-
J., M.M., F.S., L.K., I.H., M.N., P.W. and G.C. collected forest disturbance data. F.P. ran the classification models, F.C. ran
the slope instability model, Y-Y.C. and S.L. ran the ORCHIDEE model, G.F. analysed the data and wrote the manuscript with
contribution from all co-authors.

**Competing interests.** The authors declare no competing financial interests.


**Acknowledgements.** The study was funded by the Exploratory Project FOREST@RISK of the European Commission, Joint
Research Centre.

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

| Data provider | Number of records | Event type | Acquisition method |
|---|---|---|---|
| Alto Adige province forest service, Italy | 1457 | Windstorm | Aerial photointerpretation and field survey |
| AVEPA - Agenzia Veneta per i Pagamenti in Agricoltura,in collaboration with U.O. Forestale of the Veneto Region; revisited by TESAF Department, University of Padova. | 1526 | Windstorm | Aerial and satellite photointerpretation + field surveys |
| Copernicus Emergency Service | 4425 | Tornado | Aerial photointerpretation |
| Department of Cartography and Geoinformatics, Perm State University, Perm, Russia | 3056 | Windstorm | Satellite data classification [a] |
| Department of Forest Management, Geomatics and Forest Economics, Institute of Forest ResourcesManagement, Faculty of Forestry, University of Agriculture in Krakow, Poland | 321 | Windstorm | Aerial photointerpretation |
| Department of Forest Resource Planning and Informatics, Faculty of Forestry, Technical University in Zvolen, Slovakia | 14 | Windstorm | Aerial photointerpretation and field survey |
| Department of Geoinformatics, Faculty of Science, Palacky University, Czech Republic | 1175 | Windstorm | Aerial photointerpretation |
| Department of Land Change Science, Swiss Federal Institute for Forest, Snow and Landscape Research WSL, Birmensdorf, Switzerland | 64 | Windstorm | Aerial photointerpretation |
| Department of forestry Mecklenburg-Vorpommern state, Germany | 2073 | Windstorm | Aerial photointerpretation |
| Forest national service of Sweden, Sweden | 19673 | Windstorm | Semiautomatic classification [b] |
| Friuli Venezia Giulia forest service, Italy | 191 | Windstorm | Aerial photointerpretation and field survey |
| geoLAB - Laboratory of Forest Geomatics, Department of Science and Technology in Agriculture, Food, Environment and Forestry, University of Florence, Italy | 1271 | Windstorm | field survey |
| Ign-Institut National de information geographique et forestiere | 21691 | Windstorm | Aerial photointerpretation |
| Laboratory of Geomatics, Institute of Land Management and Geomatics, Aleksandras Stulginskis University, Lithuania | 14571 | Windstorm | Aerial photointerpretation |
| National Forest Centre, Forest Research Institute, Slovakia | 555 | Windstorm | Aerial photointerpretation |
| North Rhine-Westphalia forest service, Germany | 13642 | Windstorm | Aerial photointerpretation |
| Trento province forest service, Italy | 3596 | Windstorm | Aerial photointerpretation and field survey |
| University of Bucharest, Faculty of Geography, Romania | 186 | Windstorm | Aerial photointerpretation and field survey |
| University of Lorraine | 256 | Windstorm | Aerial photointerpretation |

**Table 1: List of institutions responsible of wind disturbance mapping and corresponding number of records collected and acquisition methods employed.** [a] Spatial delineation of tornado-related impacts on forests have been based on a semi-automatic algorithm and every record has been singularly validated based on visual inspection of high-resolution of satellite images (Shikhov and Chernokulsky, 2018). [b] Area subject to wind disturbances have been retrieved for FORWIND by intersection of the 2005 registered forest clear-cuts between 2005-01-07 and 2005-12-31


([http://skogsdataportalen.skogsstyrelsen.se/Skogsdataportalen/](http://skogsdataportalen.skogsstyrelsen.se/Skogsdataportalen/)) with the spatial delineation of the Gudrun storm (Gardiner et al., 2010). The use of forest clear-cuts as proxy for wind-affected areas is reasonable because the morning after the storm all normal felling activity stopped and moved to storm damaged areas (Swedish Forest Agency, personal communication).

| Class of damage | Definition of damage (D) | Degree of damage |
|---|---|---|
| **France 2009** | 0 | no forest area (not included in FORWIND) | |
| | 1 | D ≤ 20% | 0.1 |
| | 2 | 20% < D ≤ 40% | 0.3 |
| | 3 | 40% < D ≤ 60% | 0.5 |
| | 4 | 60% < D ≤ 80% | 0.7 |
| | 5 | 80% < D ≤100% | 0.9 |
| | 6 | marginally affected | missing data |
| | 7 | missing data | missing data |
| **Lithuania 2010** | 0 | no damage (not included in the FORWIND) | |
| | 1 | D ≤ 25% | 0.125 |
| | 2 | 25% < D ≤ 50% | 0.375 |
| | 3 | 50% < D ≤ 75% | 0.625 |
| | 4 | D > 75% | 0.875 |
| **Germany 2017** | 1 | D ≤ 50% | 0.25 |
| | 2 | 50% < D ≤ 90% | 0.7 |
| | 3 | 90% > D | 0.95 |
| **Italy 2018 (Trentino Alto Adige)** | 1 | D ≤ 30% | 0.15 |
| | 2 | 30% < D ≤ 50% | 0.4 |
| | 3 | 50% < D ≤ 90% | 0.7 |
| | 4 | D > 90% | 0.95 |


**Table 2: Conversion table to pass from class of damage to degree of damage.** Records of windstorms occurred in Italy in 2015 (Toscana) and in 2018 (Veneto) are already expressed as damage degree in a consistent range between 0 (no damage) and 1 (full destruction of forest pattern).

| Attribute name | Description |
| --- | --- |
| Id_poly | Identifier code |
| EventDate | Date of event (MM/DD/YYYY) |
| StormName | Storm name |
| EventType | Type of event: windstorm/tornado |
| Country | Country where the wind disturbance occurred |
| Area | Area affected by wind disturbance (in hectares) |
| Perimeter | Perimeter of the forest area affected by wind disturbance (in meters) |
| Damage_deg | Damage degree (-) |
| Methods | Acquisition method |
| Dataprovid | Data provider responsible of the wind disturbance mapping |
| Source | Original source of the data |

**Table 3: Attribute table of the FORWIND database.** Name and description of the attributes associated to each wind disturbance in FORWIND and listed in the .dbf file. Missing data are reported as -999.

| Country code | Number of records | Accumulated affected area (ha) | Median affected area (ha) | Standard deviation of affected area (ha) |
|---|---|---|---|---|
| AU | 646 | 1222.15 | 0.78 | 5.69 |
| CH | 64 | 41.28 | 0.26 | 0.79 |
| CZ | 1175 | 540.98 | 0.14 | 1.67 |
| DE | 18909 | 34075.95 | 0.64 | 5.33 |
| FR | 21947 | 875407.23 | 8.79 | 993.80 |
| IE | 561 | 541.03 | 0.36 | 1.60 |
| IT | 8041 | 33991.67 | 1.06 | 14.20 |
| LT | 14571 | 13378.80 | 0.53 | 1.28 |
| PL | 345 | 46065.34 | 24.03 | 573.29 |
| RO | 186 | 417.59 | 0.80 | 4.92 |
| RU | 3056 | 17188.38 | 0.85 | 25.41 |
| SE | 19673 | 24496.26 | 0.81 | 1.73 |
| SK | 569 | 9150.24 | 0.65 | 118.65 |
| Europe | 89743 | 1056516.91 | 1.07 | 493.20 |

**Table 4: Statistics of wind disturbance records collected in the FORWIND database aggregated at country level and for whole Europe.**

| Country code | Dates of damaging wind events recorded in FORESTORM during the 2000-2018 period | Dates of damaging wind events recorded in FORWIND | Damaging wind events recorded in FORESTORM during the 2000-2018 period and missing in FORWIND | FORWIND representativeness (-) |
|---|---|---|---|---|
| AU | 2008.01; 2008.03 | 2018.10 | 2008.01; 2008.03 | 0.333 |
| BE | 2010.02 | none | 2010.02 | |
| BG | none | none | none | |
| CH | 2002.01; 2003.01; 2004.01; 2007.01; 2008.12; 2009.01; 2009.02 | 2017.08 | 2002.01; 2003.01; 2004.01; 2007.01; 2008.12; 2009.01; 2009.02 | 0.125 |
| CY | none | none | none | |
| CZ | 2007.01; 2008.03 | 2007.01 | 2008.03 | 0.500 |
| DE | 2002.10; 2006.02; 2006.11; 2007.01; 2008.01; 2008.02; 2008.03; 2010.02 | 2007.01; 2017.11; 2018.01 | 2002.10; 2006.02; 2006.11; 2008.01; 2008.02; 2008.03; 2010.02 | 0.300 |
| DK | 2000.01; 2005.01; 2006.11; 2008.01; 2008.02 | none | 2000.01; 2005.01; 2006.11; 2008.01; 2008.02 | 0.000 |
| EE | 2005.01; 2008.02 | none | 2005.01; 2008.02 | 0.000 |
| ES | 2009.01; 2010.02 | none | 2009.01; 2010.02 | 0.000 |
| FI | 2001.unknown | none | 2001.unknown | 0.000 |
| FR | 2000.10; 2003.07; 2004.12; 2006.10; 2009.01; 2010.02; 2013.unknown | 2009.01; 2010.02 | 2000.10; 2003.07; 2004.12; 2006.10; 2013.unknown | 0.286 |
| GR | none | none | none | none |
| HR | none | none | none | none |
| HU | none | none | none | none |
| IE | 2005.01; 2014.unknown | 2014.02 | 2005.01 | 0.500 |
| IS | none | none | none | none |
| IT | none | 2015.03; 2018.10 | none | 1.000 |
| LT | 2005.01; 2008.02 | 2010-08 | 2005.01; 2008.02 | 0.333 |
| LU | 2010.02 | none | 2010.02 | 0.000 |
| LV | 2005.01; 2007.01; 2008.02 | none | 2005.01; 2007.01; 2008.02 | 0.000 |
| MT | none | none | none | none |
| NL | 2002.10; 2007.01 | none | 2002.10; 2007.01 | 0.000 |

| | | | | |
|---|---|---|---|---|
| NO | 2000.11; 2000.12; 2001.08; 2001.11; 2003.12; 2006.11; 2007.01; 2008.01 | none | 2000.11; 2000.12; 2001.08; 2001.11; 2003.12; 2006.11; 2007.01; 2008.01 | 0.000 |
| PL | 2007.01; 2008.01; 2008.02; 2008.03 | 2013.12; 2017.08 | 2007.01; 2008.01; 2008.02; 2008.03 | 0.333 |
| PT | 2010.02 | none | 2010.02 | 0.000 |
| RO | none | 2005.06 | none | 1.000 |
| RU | none | multiple tornado events | none | 1.000 |
| SE | 2001.11; 2002.01; 2003.unknown; 2005.01; 2006.11; 2007.01; 2008.01; 2008.02 | 2005.01 | 2001.11; 2002.01; 2003.unknown; 2006.11; 2007.01; 2008.01; 2008.02 | 0.125 |
| SI | none | none | none | none |
| SK | 2004.11 | 2004.11; 2014.05 | none | 1.000 |
| UK | 2000.10; 2002.10; 2005.01(.08); 2005.01(.11); 2006.11; 2007.01(.18); 2007.01(.25); 2007.06; 2007.11 | none | 2000.10; 2002.10; 2005.01(.08); 2005.01(.11); 2006.11; 2007.01(.18); 2007.01(.25); 2007.06; 2007.11 | 0.000 |
| Europe | | | | 0.626 \| 0.297 |

**Table 5: Representativeness of FORWIND.** The first estimate of representativeness at Europe level accounts for damaging wind events that occurred during the 2000-2018 period in the countries currently included in FORWIND. The second estimate of representativeness at Europe level accounts for all damaging events occurring during the 2000-2018 period, including those countries currently missing in FORWIND.

| Plant Functional Type | Model parameters | | | Coefficient of determination ($R^2$) |
|---|---|---|---|---|
| | $p_1$ | $p_2$ | $p_3$ | |
| BrDe | 0.040 (0.028, 0.052) | -0.223 (-0.279, -0.167) | 0.718 (0.662, 0.773) | 0.905 |
| BrEv | 0.051 (0.034, 0.068) | -0.265 (-0.344, -0.187) | 0.727 (0.649, 0.805) | 0.842 |
| NeDe | 0.050 (0.031, 0.070) | -0.277 (-0.367, -0.188) | 0.757 (0.668, 0.846) | 0.848 |
| NeEv | 0.025 (0.015, 0.036) | -0.157 (-0.206, -0.108) | 0.695 (0.646, 0.743) | 0.902 |


**Table 6: Parameters and performance of fitting regression models expressing the degree of damage as a function of the area affected.** The relationship between the degree of damage ($y$) and the area affected by wind disturbance ($x$) is expressed by the following general quadratic polynomial function: $y = p_1 \cdot x^2 + p_2 \cdot x + p_3$, where $p_1, p_2$ and $p_3$ are the coefficients of the equation. Coefficients are listed in the table with their 95% confidence interval in brackets. Model performance is quantified in terms of coefficient of determination ($R^2$). Models, and corresponding parameters and performance, are evaluated separately

for broadleaves deciduous (BrDe), broadleaves evergreen (BrEv), needleleaf deciduous (NeDe) and needleleaf evergreen (NeEv).


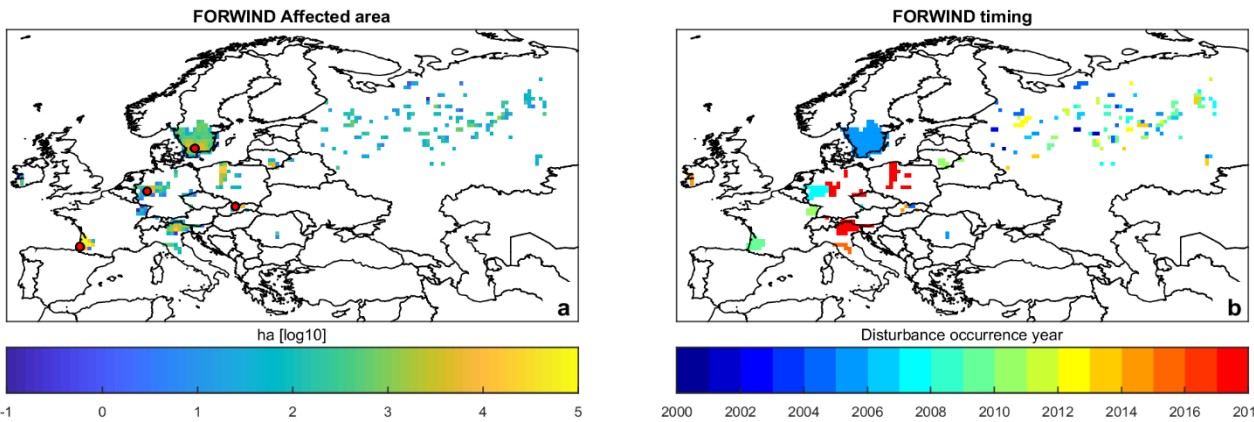

**Figure 1: Spatial and temporal distribution of wind disturbances in the FORWIND database.** (**a**) The total area affected by wind disturbances over the multi-year observational period (2000-2018) in 0.5-degree cells. (**b**) Wind disturbance occurrence year in the same cells. Red circles in (**a**) refer to site locations shown in Fig. 2.


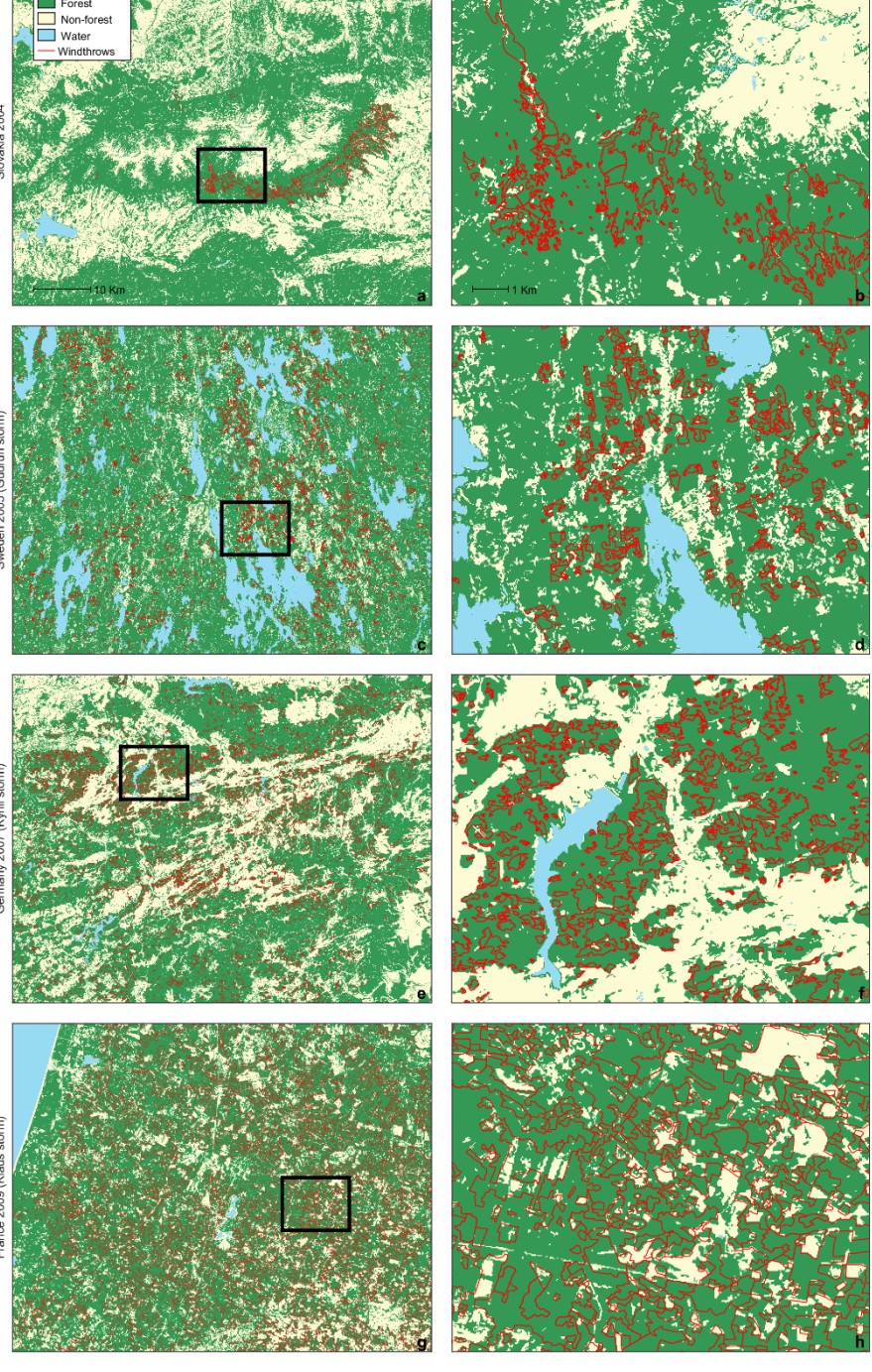

**Figure 2: Examples of wind disturbances recorded in the FORWIND database.** (**a**,**b**) Tatra Mountains, Slovakia, affected by a windstorm in 2004. (**c**,**d**) Southern Sweden affected by the Gudrun storm in 2005. (**e**,**f**) Western Germany affected by the Kyrill storm in 2007. (**g**,**h**) Western France affected by the Klaus storm in 2009. Wind disturbances recorded in the FORWIND database are shown as red polygons. Background colors show forest and non-forest areas derived from the 25-meter forest cover map of 2000 (Pekkarinen et al., 2009) while water bodies are derived from the 25-meter land cover type map of 2006 (Kempeneers et al., 2011) (https://forest.jrc.ec.europa.eu/en/past-activities/forest-mapping/#Downloadforestmaps). Site locations in (**a**,**c**,**e**,**g**) are shown in Fig. 1a whereas zoomed plots in (**b**,**d**,**f**,**h**) refer to black boxes in (**a**,**c**,**e**,**g**).

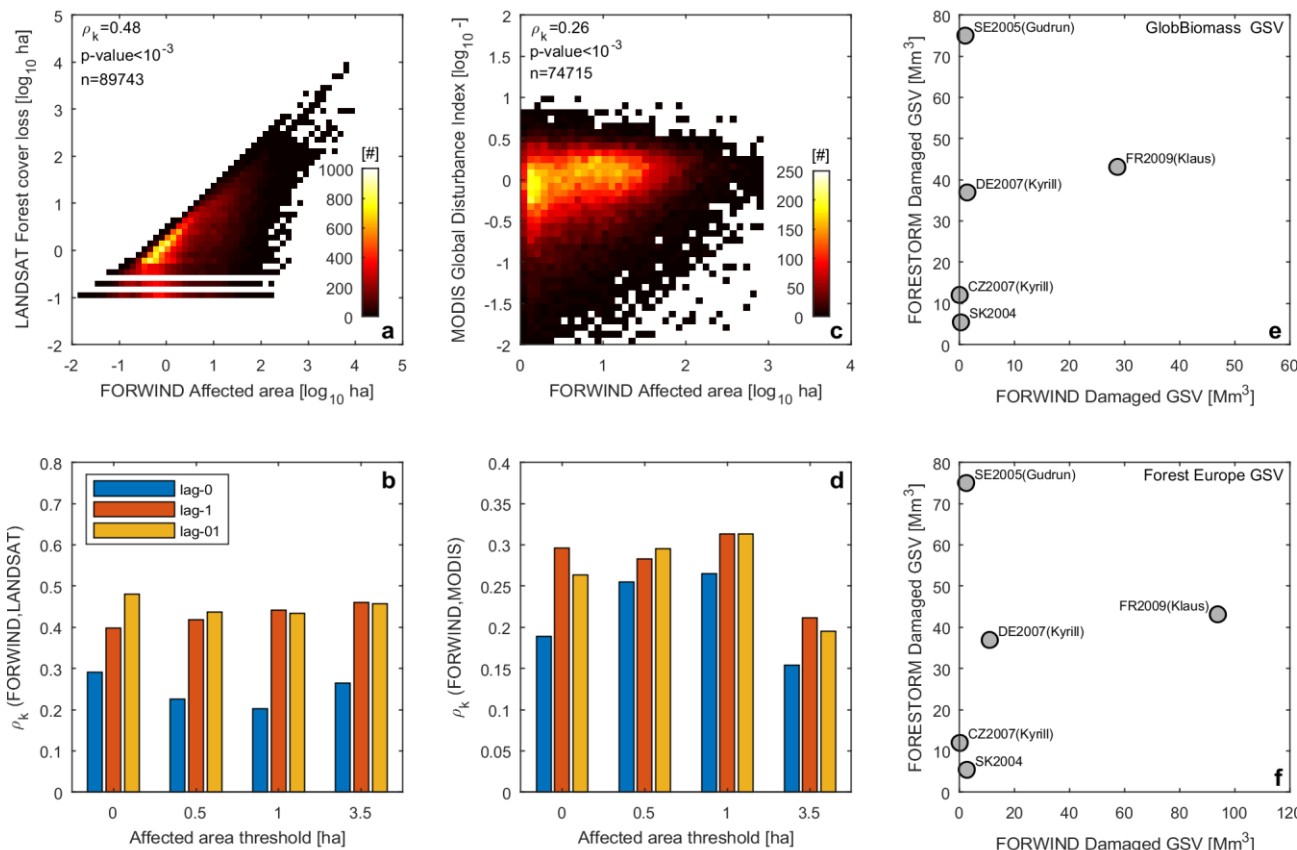


**Figure 3: Validation of the FORWIND database.** (**a**) Density plot of FORWIND affected area versus LANDSAT-derived forest cover loss, both expressed in logarithmic scale and for lag-01 effects. The color reflects the number of records, top left labels report the Spearman rank correlation coefficient ($\rho_k$), the significance (p-value) and the sample size (n). (**b**) Spearman rank correlation coefficients for different affected area thresholds (on the x-axis) and different lagged effects displayed in color bars. Lagged effects considered include the forest cover loss cumulated over the event of a given year together with that of the following year

(lag-01), forest cover loss estimated for the year event only (lag-0) and forest cover loss estimated for the following year only (lag-1). (**c**) and (**d**) as (**a**) and (**b**) but for the MODIS-derived Global Disturbance Index in place of Landsat-derived forest cover loss. (**e**) Scatter plot of damaged growing stock volume estimated from FORWIND (on the x-axis) and FORESTORM (on the y-axis) for five windstorms: Slovakia in 2004 (SK2004); Sweden in 2005 (SE2005 (Gudrun)), Germany in 2007 (GE2007 (Kyrill), the Czech Republic in 2007 (CZ2007 (Kyrill)) and France in 2009 (FR2009 (Klaus)). FORWIND estimates are derived using GlobBiomass-derived estimates of GSVs and reported damage degree information. (**f**) as (**e**) but with estimates of GSVs derived from Forest Europe national inventories and

assuming a 100% damage degree for all FORWIND records.

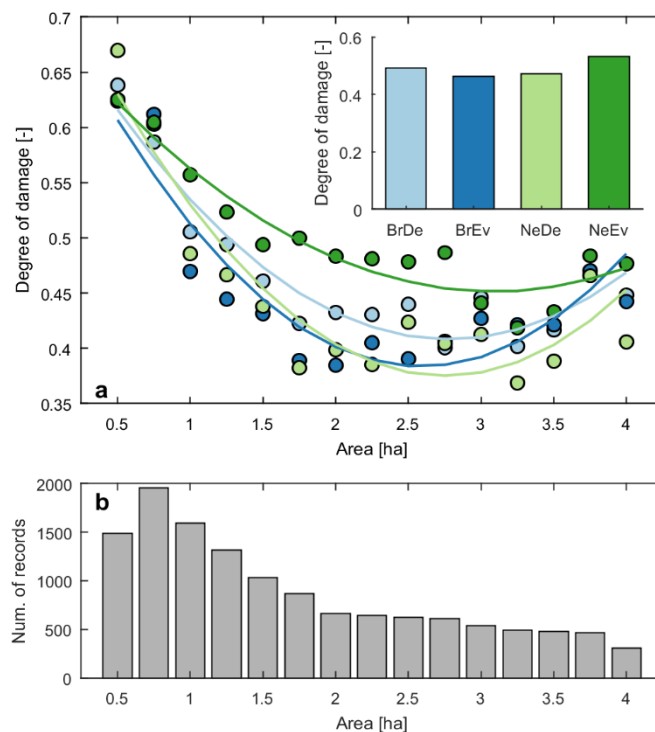

**Figure 4: Scaling relations of the degree of damage.** (a) Relation between the area affected by wind disturbance (on the x-axis) and degree of damage (on the y-axis) as derived from the FORWIND database for different PFTs, including broadleaves deciduous (BrDe), broadleaves evergreen (BrEv), needleleaf deciduous (NeDe) and needleleaf evergreen (NeEv). PFT-specific averaged values, visualized in circles of different colour, were derived using bins that spanned the sampled range and using their cover fractions as weights. The fitted quadratic polynomial functions are shown by continuous line, while their parameters and performances are reported in Table 5. The inset box shows the average degree of damage computed separately for each PFT using the whole set of records. (b) Frequency distribution of the samples (on the y-axis) over the gradient of area affected by wind disturbance (on the x-axis).



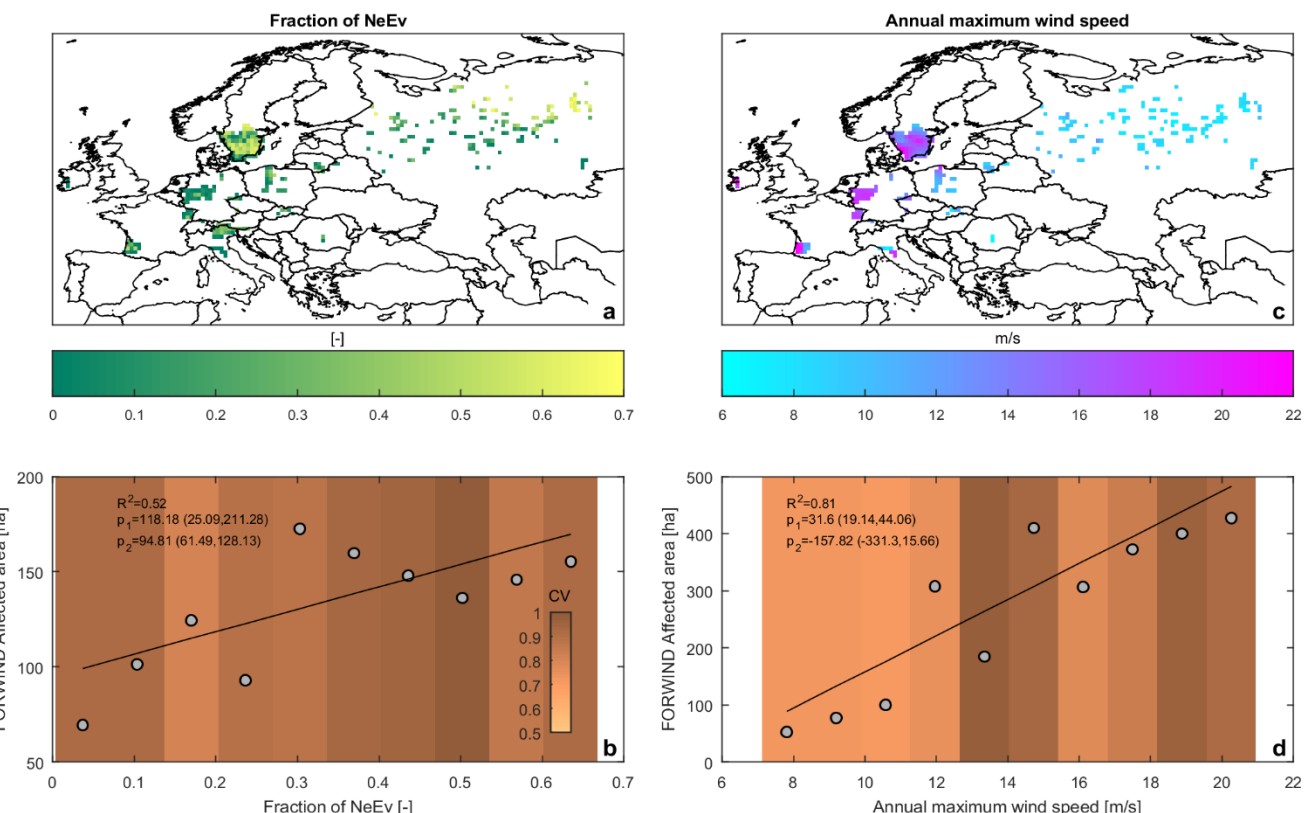

**Figure 5: Susceptibility factors and drivers of forest vulnerability to wind disturbances.** (**a**) Spatial map of the fraction of evergreen needleleaf forest (NeEv). (**b**) Relation between the fraction of NeEv (on the x-axis) and area affected by wind disturbances (on the y-axis) as derived from the FORWIND database. Averaged values, shown in grey circles, were derived using bins that spanned the sampled range. Colour patterns reflect the coefficient of variation within each bin. The fitted linear regression model is shown in black line with the coefficient of determination ($R^2$), slope ($p_1$) and intercept ($p_2$) reported in the labels. The 95% confidence interval for each of the coefficient is shown in brackets. (**c**) Spatial map of annual maximum wind speed; (**d**) as (**b**) but for annual maximum wind speed in place of the fraction of NeEv. The grid cells in (**a**) and (**c**) with no wind disturbances occurred over the 2000-2018 period are masked out.

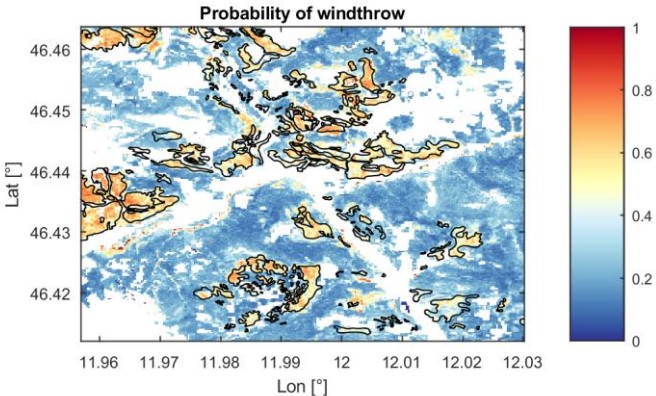

**Figure 6: Remote sensing classification of windthrows.** Probability of windthrow obtained from random forest classification of Sentinel-2 reflectance bands and their tasselled cap transformation in a sampled area of the Dolomites Mountains in Northern Italy affected by the Vaia storm of October 2018. Black polygons show the actual wind disturbances.

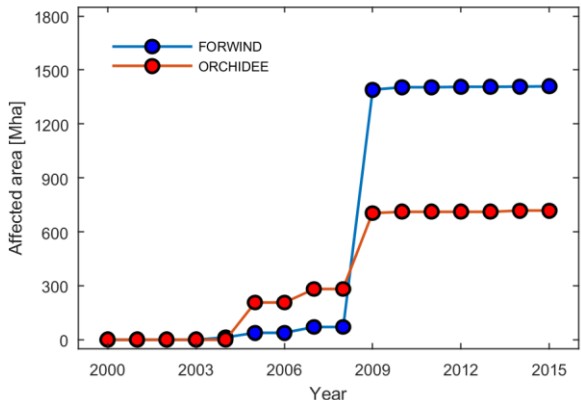

**Figure 7: Observed and simulated cumulated forest area damaged by windstorms between 2000 and 2015 over Europe.** The observed damage area was extracted from the FORWIND dataset (shown in blue) whereas the simulated area comes from ORCHIDEE r4262 with $R_f=6$ (shown in red).

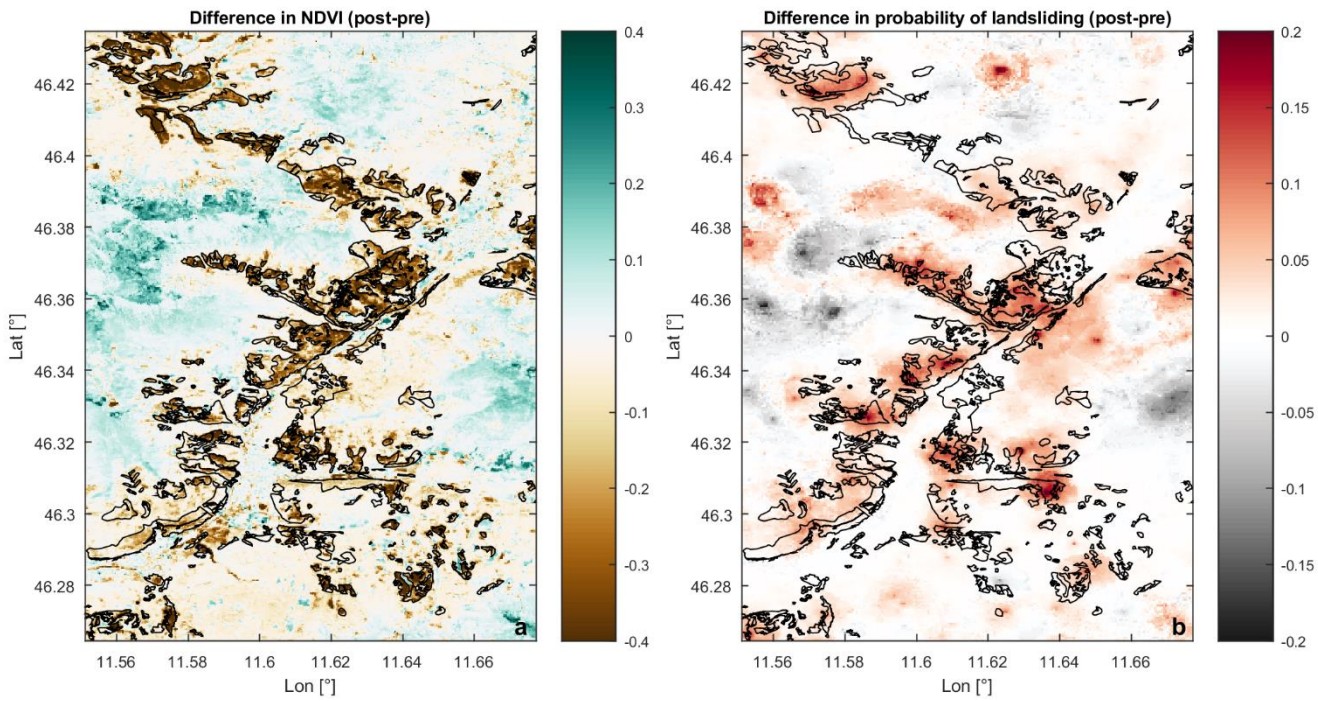

**Figure 8: Analysis of the indirect effects of wind damages on slope instability.** Changes in NDVI and probability of landsliding following the Vaia storm of October 2018 in the Dolomites Mountains in Northern Italy, in (**a**) and (**b**), respectively. Black polygons show the actual wind disturbances.