# Peer review of "A spatially-explicit database of wind disturbances in European forests over the period 2000-2018"

_Earth System Science Data, 2019_

## Referee Comment (RC1) · Rupert Seidl (Referee) · 17 Oct 2019

Forzieri et al. present the first spatially explicit collection of forest areas disturbed by wind in Europe. This is a highly timely and important effort, as natural disturbances are increasing in Europe, yet we largely lack high quality datasets for understanding and modeling these processes. Compilations such as the one presented here are thus the prerequisite for improving our predictive capacity of natural disturbances.

The current dataset follows a data compilation approach, i.e. previous records from a variety of different sources are combined in a single database. The authors thus synthesize a number of past regional efforts and make them available for the scientific

community. I overall find this work to be highly relevant and useful, and commend the authors for their efforts.

I also appreciated the comparison of the dataset against estimates derived from Landsat, Modis, and grey literature. However, I would not call this an evaluation or validation of the current dataset, as all these data are derived differently, pertain to different resolutions, and apply different thresholds for recording a disturbance, so it is basically comparing apples to oranges. If anything, I belief the current data to be the most accurate of all the datasets compared, and deviations between the products are largely the effect of differences in methodology (I would assume that also Landsat and Modis have a moderate correlation at best). This for me underlines the importance of ground-true datasets as the one presented here.

I find that two things currently limit the utility of the dataset though, and would suggest that these aspects could still be improved in a moderate revision of the manuscript before publication. First, the threshold severity that was applied in the assessments compiled here is not defined. This means that the polygons compiled here could have anything from 1% to 100% of the trees thrown or broken by wind. This ambiguity strongly limits the utility of the data for ecological analyses. It seems from the text that severity measures are available for at least some of the polygons, and I suggest that you also include them in the data where you have them.

Second, while the sampling via a PubMed and Scopus search is clearly described, it remains unclear how representative the compiled polygons are for the wind disturbances that occurred within a year in a given country. Looking at Table 4, I for instance wonder whether the 64 polygons on record for Switzerland are the total forest area that was affected by wind in this country, or whether this is a (random?) sample of all areas affected by wind. Again, information on the representativeness of the sample would be important for making ecological inference. As for my previous point, I have the feeling from reading the text that you have an understanding of how representative your database is for at least some countries and storm events. Adding this type of

information would certainly increase the value of your dataset for the further analyses.

Overall, I find this to be a highly relevant dataset, and recommend publishing it after moderate revisions. Some more minor comments are below:

l59: their excess. . . meaning unclear

l66, l70, and many other instances throughout the text: a space is missing before the parenthesis

l69: of the average annual harvest rate. . . where? in all of Europe? in the effected countries? Be more specific here. The same applies to a similar statement in line 70.

l78: substitute "Europe" for "European"

l80: not true for Senf et al. (2018), which is based on satellite information as far as I recall

l86: Full stop is missing after "decades"

l104: regardless of the degree of damage: Does this mean that it was enough for a single tree to fall within a 100 ha tract for the area to be admitted to your database?

l133: impressive!

l135: forest disturbance patch

l243: one issue that I see there (that also might account for the differences you find) is: If you use ForestEurope values for GSV these are the averages per country. However, wind disturbances are predominately affecting older stand and more productive sites (as both have taller trees), which means that the actual GSV of areas affected by wind might be considerably higher than the country-level averages.

l268-270: I don't fully understand this

l276-277: I don't agree with this statement (think about Abies alba or Pinus sylvestris); I think it is mainly the prevalence of Picea abies that drives the relationship (for which

the statement you give is correct).

Figure 1: Can you put the units next to the scale bar, rather than in the figure header?

[Figure]

---

## Referee Comment (RC2) · Anonymous Referee #2 · 21 Oct 2019

General Comment: This study integrated the windthrow observations from aerial photointerpretation and field survey and compared the results with remote sensing indexes and total damaged wood reported in the FORESTORMS database. Their work provides a specially-explicated storm-affected area which is helpful to improve the modeling framework on simulating storm damage in the Earth system model. The damage rate within a storm-affected area can be also found in this data synthesis. However, I could not access any further information about this information. I found that it is very important to reveal the relationship between the degree of damage and affected area among various tree species, such as needle-leaved forests or broadleaf forests, from the model development point of view. I thus recommended that the authors report the

relationship between the damage rate and storm-affected area in this dataset. Along with this discussion, the authors may/can introduce the section of data comparison by analyzing their dataset and other remote sensing indexes by using different thresholds for accessing, justifying, or distinguishing the windthrow damage.

The work made by the authors is not trivial and I support the publication of this study in ESSD. Before publishing this work, I have a few specific comments listed below:

1. P5L435L: Please explain the reason for using a 500 m2 clear cut area to identify the wind damage due to Gudrun in 2005. Besides, the storm Gudrun caused a super huge damage area which required several years to clean the damaged forests.

2. P8L248: The authors argue that a possible reason for underestimating the damaged wood volume/biomass may due to the uncertainty of initial biomass within the FORWIND identified the storm-affected area. The authors should provide the number of mean biomass for the FORWIND identified storm-affected area. Otherwise, I think the uncertainty for estimating the damaged wood volume/biomass due to windthrow might originate from missing interpretation of aerial photos.

3. P10L299: Please check the citation of the study made by Bonan and Doney (2018) for the implementation of a windstorm effect in land surface models.

4. Please add a space between texts and parentheses.

---

## Author Comment (AC1) · 28 Nov 2019

First of all, we would like to thank the referee for the insightful and constructive comments. In our revised version of the manuscript we tried to address all his comments and suggestions in order improve the robustness of the analysis and the clarity of the interpretation.

In the following, we respond to each reviewer's comment by referring to line numbers of the revised non-tracked version, when not differently indicated.

**Reviewer 1, Rupert Seidl**

*Forzieri et al. present the first spatially explicit collection of forest areas disturbed by wind in Europe. This is a highly timely and important effort, as natural disturbances are increasing in Europe, yet we largely lack high quality datasets for understanding and modeling these processes. Compilations such as the one presented here are thus the prerequisite for improving our predictive capacity of natural disturbances.*

*The current dataset follows a data compilation approach, i.e. previous records from a variety of different sources are combined in a single database. The authors thus synthesize a number of past regional efforts and make them available for the scientific community. I overall find this work to be highly relevant and useful, and commend the authors for their efforts.*

> We thank the reviewer for his positive comment. We have revised the text in light of it. Please, note that, inspired from some comments received from rev2, we decided to expand in the revised version the potential applications of FORWIND encompassing several challenging topics and scientific fields including forest vulnerability modelling, scaling relations of wind damages, remote sensing monitoring of forest disturbances, representation of uproot and break trees in large-scale land surface models and hydrogeological risks associated to wind disturbances. We believe that this new material further improves the manuscript and may facilitate the use of FORWIND in multiple scientific disciplines and contexts.

*I also appreciated the comparison of the dataset against estimates derived from Landsat, Modis, and grey literature. However, I would not call this an evaluation or validation of the current dataset, as all these data are derived differently, pertain to different resolutions, and apply different thresholds for recording a disturbance, so it is basically comparing apples to oranges. If anything, I belief the current data to be the most accurate of all the datasets compared, and deviations between the products are largely the effect of differences in methodology (I would assume that also Landsat and Modis have a moderate correlation at best). This for me underlines the importance of ground-true datasets as the one presented here.*

> We agree with the reviewer's comment. Indeed, a standard validation exercise of FORWIND is not possible due to the lack of alternative datasets with similar spatially explicit representation of wind disturbances.

> **Action taken:**

➔ We have changed the heading of section 4.1. from "Technical validation" to "Comparison of FORWIND with satellite-based metrics and national inventories".

1. *I find that two things currently limit the utility of the dataset though, and would suggest that these aspects could still be improved in a moderate revision of the manuscript before publication. First, the threshold severity that was applied in the assessments compiled here is not defined. This means that the polygons compiled here could have anything from 1% to 100% of the trees thrown or broken by wind. This ambiguity strongly limits the utility of the data for ecological analyses. It seems from the text that severity measures are available for at least some of the polygons, and I suggest that you also include them in the data where you have them.*

We agree with the reviewer's comment on the importance of including information about disturbance severity. However, we believe that the reviewer may have overlooked this information, as it is already included in our database (see attribute "Damage_degree").

A damage classification for forest disturbances was originally recorded for windstorms that occurred in France in 2009, in Lithuania in 2010, in Germany in 2017, in Italy in 2015 and –for part of the records - in 2018. In order to make these records comparable in terms of the severity of damage, the original classes were harmonized into a single damage metric following the rationale reported in Table 2. The resulting severity (or degree of damage) varies in a consistent range between 0 (no damage) and 1 (full destruction of forest patch). Missing data for the remaining wind disturbances are reported as -999. The harmonization of the degree of damage was already described in our previous submission at lines 121-124 and table 2. The database includes a specific attribute named "Damage_degree" (see also Table 3) in which the severity (or degree of damage) is reported.

We did not apply any severity threshold in our data collection for two key reasons. First, information on the degree of damage is reported only for a part of the database (~48%). While we agree with the reviewer that the degree of damage is key information for ecological analyses, we also believe that wind disturbances can be meaningfully characterized and analyzed when damage levels are not recorded. Second, the definition of a threshold to include/exclude records based on their degree of damage would necessarily imply a subjective choice, potentially questionable depending on the use of the data and the question addressed by the ecological analysis. Based on the above-mentioned considerations, we opted to include all records in FORWIND and report the degree of damage when available. In our opinion, this approach does not limit the use of the database but allows the user to set severity thresholds appropriate for her or his specific tasks.

**Action taken:**

➔ We have clarified this in the revised text. We hope that the reviewer agrees on this strategy.

➔ Furthermore, following a comment from reviewer 2, we explored in the revised version the scaling relations of degree of damage across different plant functional types.

2. *Second, while the sampling via a PubMed and Scopus search is clearly described, it remains unclear how representative the compiled polygons are for the wind disturbances that occurred within a year in a given country. Looking at Table 4, I for instance wonder whether the 64 polygons on record for Switzerland are the total forest area that was affected by wind in this country, or whether this is a (random?) sample of all areas affected by wind. Again, information on the representativeness of the sample would be important for making ecological inference. As for my previous point, I have the feeling from reading the text that you have an understanding of how representative your database is for at least some countries and storm events. Adding this type of information would certainly increase the value of your dataset for the further analyses.*

We agree with the reviewer's comment. The overall aim of the study is to develop a database of forest disturbances that is as comprehensive as possible. To this aim 26 research institutes and forestry services from different European countries were involved in the data collection. The database includes all major windstorms occurred over the observational period (e.g., Gudrun, Kyrill, Klaus, Xhynthia and Vaia). Despite such unique joint effort (89,743 records have been collected in this first release), we recognize that FORWIND could miss some wind damage occurrences, as also explicitly mentioned on lines 245-246. For this reason, further data contributions are encouraged in order to continuously update and improve FORWIND (lines 306-307).

Evaluating quantitatively the degree of representativeness of FORWIND is very challenging because the known wind events may represent only a fraction of the overall occurrences. Wind disturbances may remain unknown for a long time. On the other hand, we agree with the reviewer's comment on

the importance of providing an estimate of the representativeness of FORWIND. This information, may also serve to drive more effectively future efforts to populate the database.

**Action taken:**

➔ According to the institutions responsible for the data acquisition, the wind disturbances recorded in FORWIND exhaustively represent the damaged forest areas caused by those specific events. However, some known damaging wind events are currently missing in the database. In order to provide a more comprehensive assessment of the representativeness of FORWIND, we derived for each country the ratio between the number of sampled wind events and the number of all wind events occurred which are known to have caused forest damages. The number of known damaging events is derived by summing up the number of distinct events recorded in FORESTORM and FORWIND during the 2000-2018 period. Therefore, the temporal representativeness ranges between 0 (all known wind disturbances are missing in FORWIND) and 1 (all known wind disturbances are included in FORWIND). Estimates of representativeness ranges between 0.13 and 1 amongst the countries included in FORWIND, with average value of 0.64 at Europe level (new table added in the revised version). However, when also countries currently missing in FORWIND are accounted for the average representativeness decreases to 0.37. These values should be viewed with caution as the estimated number of total damaging wind events resulting from FORWIND and FORESTORM could likely deviate from the actual ones. Future efforts should be aimed to populate FORWIND with those damaging wind events actually missing.

➔ We have described the representativeness metric in the revised version of the manuscript and added a dedicated new table. We also recall the representativeness of FORWIND in the abstract.

*Overall, I find this to be a highly relevant dataset, and recommend publishing it after moderate revisions. Some more minor comments are below:*

In the following lines, we tried to address all the remaining issues.

**Minor comments**

*l59: their excess... meaning unclear*

**Action taken:**

➔ We have rephrased with "occurrence".

*l66, l70, and many other instances throughout the text: a space is missing before the parenthesis*

The issue was due to the setup of the plug-in used for citations and bibliography.

**Action taken:**

➔ We have fixed the problem in the revised version.

*l69: of the average annual harvest rate... where? in all of Europe? in the effected countries? Be more specific here. The same applies to a similar statement in line 70.*

**Action taken:**

➔ We have rephrased the statements and the percentages now refer to the corresponding countries affected. Percentage values are retrieved from official roundwood statistics, used here as a proxy of harvest, reported in the FAOSTAT database.

*l78: substitute "Europe" for "European"*

**Action taken:**

➔ According to the reviewer's comment, we have corrected the text.

*l80: not true for Senf et al. (2018), which is based on satellite information as far as I recall*

Senf et al. (2018), amongst a series of other data sources, utilized country-scale estimates of natural disturbances reported in previous publications (Schelhaas et al., 2003; Seidl et al., 2014). However, we recognize that the mentioned article has implemented a sophisticated approach mostly based on satellite data and where country scale estimates are only partially exploited. Therefore, in agreement with the reviewer's comment, we agree that it may be not fully appropriate to cite Senf et al. in this context.

**Action taken:**

➔ We have removed the citation in the revised version of the manuscript.

*l86: Full stop is missing after "decades"*

**Action taken:**

➔ We have corrected the typo.

*l104: regardless of the degree of damage: Does this mean that it was enough for a single tree to fall within a 100 ha tract for the area to be admitted to your database?*

Each polygon represents the spatial delineation of the forested area affected by wind damage (uprooted and broken trees). Following the example hypothesized by the reviewer, the area of the polygon where only a single tree felt, will reflect the approximate area covered by such single tree, surely much lower than 100 ha. Consider that the acquisition of the polygons was made by aerial photointerpretation or field survey. Therefore, the polygons are delineated when a reasonably homogeneous patch of damaged forest is detected form the ground or remotely. As detailed in the response to your comment #1, we intentionally avoided to fix thresholds on the degree of damage and areal extent of affected forested patches. It is up to the user to decide what screening to implement based on their specific purpose.

**Action taken:**

➔ We have further clarified this concept in the revised version of the manuscript.

*l133: impressive!*

Thank you! We are considering to implement FORWIND in a web portal complemented by a dedicated tool to automatically integrate and check new data acquisitions.

*l135: forest disturbance patch*

**Action taken:**

➔ According to the reviewer's suggestion, we have corrected the text.

*l243: one issue that I see there (that also might account for the differences you find) is: If you use ForestEurope values for GSV these are the averages per country. However, wind disturbances are predominately affecting older stand and more productive sites (as both have taller trees), which means that the actual GSV of areas affected by wind might be considerably higher than the country-level averages.*

We agree with the reviewer.

**Action taken:**

➔ In order to account for the presence of typically more productive forests in areas affected by wind disturbances, Forest Europe-derived GSVs were rescaled based on the ratio between the average tree height computed over the wind-affected areas and the average tree height computed over all vegetated lands in the country. In such simplified approach, we implicitly assume a linear relation between GSV and tree height. Tree height values where retrieved from 1-km spaceborne light detection and ranging (lidar) data acquired in 2005 by the Geoscience Laser Altimeter System (GLAS) aboard ICESat (Ice, Cloud, and land Elevation Satellite), (https://webmap.ornl.gov/wcsdown/dataset.jsp?ds_id=10023) (Simard et al., 2011). Results are largely consistent with our previous estimates, yet the discrepancies between estimates derived from FORESTORMS and FORWIND are slightly lower than before. The tree height-based rescaling factors ranges between 0.8 and 1.24, with value lower than 1 only for the event Klaus occurred in France in 2009. We have noted that in our previous estimates we used the wrong damaged GSV for the Gudrun event. Now, numbers are correct. We have described the afore-mentioned method in the revised version and updated figure 3.

**Action taken:**

➔ We have rephrased the sentence as follows: "The high spatial variability of the considered metrics and the potential effects of additional environmental factors not considered in this exercise may potentially mask the functional relations between response variable and predictors. In order to reduce such potential sources of noise, response variables and predictors were spatially averaged over the sampled range of the predictors (bin sizes of 10% and 2 m/s for fraction of ENF and annual maximum wind speed, respectively)."

*l276-277: I don't agree with this statement (think about Abies alba or Pinus sylvestris); I think it is mainly the prevalence of Picea abies that drives the relationship (for which the statement you give is correct).*

We thank the reviewer for this comment. We agree.

**Action taken:**

➔ We have modified the sentence in the revised version as follows: "The emerging relation is likely driven by the relatively high abundance of Picea Abies in the sampled forest areas. Indeed this tree species is typically characterized by shallower rooting systems than other common species."

*Figure 1: Can you put the units next to the scale bar, rather than in the figure header?*

**Action taken**

➔ We have modified the figure according to the reviewer's suggestion. For consistency, we have also modified Figure 5.

---

## Author Comment (AC2) · 28 Nov 2019

First of all, we would like to thank the referee for the insightful and constructive comments. In our revised version of the manuscript we tried to address all hei/his comments and suggestions in order improve the robustness of the analysis and the clarity of the interpretation.

In the following, we respond to each reviewer's comment by referring to line numbers of the revised non-tracked version, when not differently indicated.

**Reviewer 2**

*General Comment: This study integrated the windthrow observations from aerial photointerpretation and field survey and compared the results with remote sensing indexes and total damaged wood reported in the FORESTORMS database. Their work provides a specially-explicated storm-affected area which is helpful to improve the modeling framework on simulating storm damage in the Earth system model.*

We thank the reviewer for her/his positive comment. Inspired by your comments (3 and 4), we decided to expand in the revised version a series of potential applications of FORWIND. They include challenging topics such as forest vulnerability modelling, scaling relations of wind damages, remote sensing-based monitoring of forest disturbances, representation of uproot and break trees in large-scale land surface models and hydrogeological risks associated to wind disturbances. We believe that this new material further improves the manuscript and may facilitate the use of FORWIND in multiple scientific disciplines and contexts.

1. *The damage rate within a storm-affected area can be also found in this data synthesis. However, I could not access any further information about this information. I found that it is very important to reveal the relationship between the degree of damage and affected area among various tree species, such as needle-leaved forests or broadleaf forests, from the model development point of view. I thus recommended that the authors report the relationship between the damage rate and storm-affected area in this dataset.*

   According to the reviewer's comment, we have explored the relationship between the degree of damage and affected area for different plant functional types.

   **Action taken:**

   ➔ In order to evaluate the relationship between the degree of damage and affected area, we estimated, for each record, the cover fractions of different plant functional types (PFTs) including broadleaves deciduous (BrDe), broadleaves evergreen (BrEv), needleleaf deciduous (NeDe) and needleleaf evergreen (NeEv). Cover fractions were retrieved from the annual land cover maps of the European Space Agency's Climate Change Initiative (ESA, 2017) (ESA-CCI, https://www.esa-landcover-cci.org/). Degrees of damage of each record are then spatially averaged over the sampled interquartile range of affected areas using a bin size of 0.25 ha. The spatial averages are computed separately for each PFTs utilizing their cover fractions as weights. Quadratic polynomial functions are finally used to fit the observations and retrieve the relationship between the degree of damage and affected area for the considered PFTs.

   Results show that all considered PFTs have generally higher degree of damage for wind disturbance with small spatial extent (Fig. 4a). This may reflect a better delineation of small affected areas when the damage are typically higher and homogeneous. Furthermore, the emerging declining scaling relations could suggest a potential dampening effect of wind severity thanks to a higher landscape heterogeneity in large areas compared to more homogeneous patterns in small forest patches.

   Model fitting shows reasonably good performances with $R^2$ ranging between 0.84 and 0.9 across the PFTs (Table 6). NeEv have generally higher degree of damage compared to the other PFTs. For this biome, the emerging relationship between the degree of damage and affected area is characterized by a prevalent quasi-monotonic pattern. The relationships found for the other PFTs show a stronger link

between degree of damage and affected area compared to NeEv, particularly over the range with larger samples (affected areas < 2 ha, Fig. 4b) as visualized by steeper slopes of the fitting functions. For BrDe, BrEv and NeDe a prominent parabolic pattern emerges distinctly driven by records with large spatial extents and relatively high degree of damage.

We stress that the above example is an oversimplification of the relationships observed in nature. More sophisticated fitting functions and more objective metrics of severity could be employed to better capture the scaling relations of the degree of damage. Therefore, the approach described should not be considered as a reference methodology but only as an informative application to explore the usefulness of the FORWIND database.

We have described the above-mentioned method and results in the revised version and added one new figure and one new table to synthesize main findings and list model parameters and fitting performance.

➔ Furthermore, we have included in the revised version of the manuscript, an example on how FORWIND can be used to improve the parameterization of land surface models in representing wind disturbances.

2. *Along with this discussion, the authors may/can introduce the section of data comparison by analyzing their dataset and other remote sensing indexes by using different thresholds for accessing, justifying, or distinguishing the windthrow damage.*

We agree with the reviewer's comment on the potential of remote sensing data to detect/attribute wind disturbances as well as to quantify the corresponding forest damages.

Previous attempts to detect and attribute wind disturbances from remote sensing data were mostly hampered by the limited number of sampled wind-affected areas available for training/testing classification algorithms. In this respect, FORWIND – given the high number of records – represent a unique source to improve classification performances over large scales and quantify wind impact. For instance, FORWIND could be used to evaluate what remote sensing indexes (or other auxiliary features) and thresholds are more appropriate to identify wind disturbances and assess their damages. Such applications, however, pose a series of challenges. Distinguishing the changes in spectral signature due to wind disturbances from those driven by human forest management or quantifying the spatial and temporal dynamics of exposed biomass are just a couple of critical issues that should be addressed in order to retrieve reasonable estimates.

**Action taken:**

➔ We have included a new section reporting an example of the use of FORWIND to classify wind-affected areas. The presented approach in the revised version should not be considered as reference methodology but as example of a potential application of FORWIND. We believe that the development of more dedicated modelling frameworks are out of the scope of this work. The major novelty of our analysis consists in having collected and harmonized more than 80,000 forest areas damaged by wind into a consistent Pan-European geospatial dataset. This is the result of a unique joint effort of 26 research institutes and forestry services across Europe. We provide FORWIND as a freely accessible product to the scientific community. We leave to the potential user the opportunity to design and develop appropriate classification tools and assessments of wind disturbances. We hope that the reviewer understand our point of view.

*The work made by the authors is not trivial and I support the publication of this study in ESSD. Before publishing this work, I have a few specific comments listed below:*

We thank the reviewer for her/his positive comments. In the following lines, we have tried to respond to its remaining comments.

3. *P5L435L: Please explain the reason for using a 500 m2 clear cut area to identify the wind damage due to Gudrun in 2005. Besides, the storm Gudrun caused a super huge damage area which required several years to clean the damaged forests.*

Aerial photointerpretation or field survey aimed to specifically delineate wind disturbances associated to Gudrun are not available. However, the use of forest clear-cuts as proxy for wind-affected areas is reasonable because the morning after the storm all normal felling activity stopped and moved to storm damaged areas (Swedish Forest Agency, personal communication). Therefore, area subject to wind disturbances recorded in FORWIND have been retrieved by intersection of the 2005 registered forest clear-cuts between 2005-01-07 and 2005-12-31 larger than 500 $m^2$ with the spatial delineation of the Gudrun storm (Gardiner et al., 2010). The initial fixed threshold of 500 $m^2$ was chosen because that value represented the threshold for which forest owners are obliged by law to communicate any clear-cuts. After a closer investigation with the Swedish Forest Service, we decided to remove such threshold in our database (revised version), in order to include smaller areas affected by wind.

4. *P8L248: The authors argue that a possible reason for underestimating the damaged wood volume/biomass may due to the uncertainty of initial biomass within the FORWIND identified the storm-affected area. The authors should provide the number of mean biomass for the FORWIND identified storm-affected area. Otherwise, I think the uncertainty for estimating the damaged wood volume/biomass due to windthrow might originate from missing interpretation of aerial photos.*

We believe that the reviewer may have misunderstood our validation exercise. We try to clarify the rationale, by referring to the two experiments reported in the text (a and b in the following lines).

a) In a first experiment, we derived, for each of the events considered, estimates of damaged GSV using the GlobBiomass dataset (Santoro et al., 2018). Such values are derived under two different scenarios: 1) accounting for the record-specific degree of damage, and 2) assuming 100% degree of damage for all records. Such values are then compared with damaged GSVs reported in FORESTORMS. This comparison shows a substantial underestimation of GSVs in FORWIND compared to FORESTORMS estimates (Fig. 3e). We pointed out that "any deviations of the mapped GSV from the true forest state are inherently translated into our damaged GSV estimates". Therefore, any errors in the GlobBiomass product are reflected in our estimates of damaged GSVs. In particular, the GSV map refers to the year 2010, therefore it is very likely that it largely reflects the biomass conditions following, rather than preceding, the windstorm events (all the five events considered in this validation exercise occurred before 2010).

b) In order to solve the above-mentioned issue, we performed an additional validation exercise. To this aim, we derived country-scale estimates of average GSVs for the year 2000 (pre-event conditions) from the State of Europe's Forest (FOREST EUROPE, 2015). We then derived the damaged GSVs by rescaling Forest Europe-derived GSVs based on the area affected by wind disturbances (from FORWIND) and the tree height in such areas (please, note the integration of tree height to incorporate a comment from reviewer 1). For such estimates of damaged GSVs we assumed 100% degree of damage. Finally, these damaged GSVs are compared with those estimates derived from FORESTORM as in the previous exercise. As we assume a 100% degree of damage, damaged GSVs reported over the x-axis in Fig. 3f reflect exactly the mean biomass located in those areas affected by wind disturbances. Therefore, the information requested by the reviewer is already reported in our results of the second experiment (b). Results of this experiment are largely in agreement

with previous estimates and show a substantial underestimation of damaged GSVs in FORWIND compared to FORESTORMS estimates. We recognize that FORWIND could miss some wind damage occurrences for instance due to incorrect detection of wind disturbances from aerial photointerpretation, as correctly pointed out by the reviewer, or difficulties to map inaccessible areas through ground survey. However, according to the institutions responsible for the data acquisition, the forest areas affected by the windstorm events considered in this validation exercise were exhaustively mapped. We therefore argue that a possible source of error may be associated to the FORESTORM database. Estimates of forest damages from FORESTORM originate from different sources and are collected by multiple actors. Hence, the loss figures should be viewed in light of their potential biases, including a possible overestimation of the true impacts.

**Actions taken:**

➔ We have clarified this in the revised version.

➔ As already mentioned, according to the institutions responsible for the data acquisition, the wind disturbances recorded in FORWIND exhaustively represent the damaged forest areas caused by those specific events. However, some known damaging wind events are currently missing in the database. Such missing events do not affect the validation exercise shown in figure 3. However, in order to provide a more comprehensive assessment of the representativeness of FORWIND, we derived for each country the ratio between the number of sampled wind events and the number of all wind events occurred which are known to have caused forest damages. The number of known damaging wind events is derived by summing up the number of distinct events recorded in FORESTORM and FORWIND during the 2000-2018 period. Therefore, the temporal representativeness ranges between 0 (all known wind disturbances are missing in FORWIND) and 1 (all known wind disturbances are included in FORWIND). Estimates of representativeness range between 0.13 and 1 among the countries included in FORWIND, with an average value of 0.67 at Europe level (see table 5). However, when also countries currently missing from FORWIND are accounted for the average representativeness decreases to 0.37. These values should be viewed with caution as the estimated number of total damaging wind events resulting from FORWIND and FORESTORM could likely deviate from the effective ones. Future efforts should be aimed to populate FORWIND with those damaging wind events actually missing. This has been described in the revised version and a dedicated table has been added (Table 5).

5. *P10L299: Please check the citation of the study made by Bonan and Doney (2018) for the implementation of a windstorm effect in land surface models.*

   **Action taken:**

➔ We have removed the referenced study and cited later in a more appropriate context.

6. *Please add a space between texts and parentheses.*

   The issue was due to the setup of the plug-in used for citations and bibliography.

   **Action taken:**

➔ We have fixed the problem in the revised version.